# Cooperative Multimodal Energy-based Model with MCMC Revision

## Abstract

This paper studies the learning problem of the energy-based models (EBM) for multimodal data. Learning EBMs via maximum likelihood estimation (MLE) typically involves Markov Chain Monte Carlo (MCMC) sampling, such as Langevin dynamics; however, noise-initialized Langevin dynamics is often ineffective and hard to mix. More critically, multimodal data contains complex inter-modal dependencies (i.e., relationships shared across modalities), making informative and coherent initializations across multimodalities particularly crucial for multimodal EBM sampling and learning. Notably, Multimodal VAEs, consisting of a shared generator model and a joint inference model, have made progress in capturing such inter-modal dependencies. But, both the shared generator and joint inference models are modelled as unimodal Gaussian (or Laplace), which can be limited in statistical expressivity for complex data and generator posterior distributions. In this work, we investigate the learning problem of the multimodal EBM, shared generator, and joint inference model by interweaving their MLE updates with respective MCMC revisions. With MCMC EBM revision, the shared generator learns to produce coherent multimodal initializations for EBM sampling. The joint inference model provides informative latent initializations as guided by MCMC posterior sampling. Both models serve as complementary initializer models that facilitate effective EBM sampling and learning, leading to realistic and coherent multimodal EBM samples. Extensive experiments demonstrate superior performance for multimodal synthesis quality and coherence compared to various baselines. Analysis, ablation studies, and supplementary experiments further validate the effectiveness and scalability of the proposed multimodal framework.

## 1 Introduction

Deep generative models (DGMs) have achieved remarkable success in modeling complex data distributions for single modalities (Ho et al., 2020; Karras et al., 2020; Hoffman, 2017; Taniguchi et al., 2022). In recent years, these advances have rapidly extended to *single-flow* multimodal frameworks (e.g., text-to-image) (Ramesh et al., 2022; Alayrac et al., 2022; Li et al., 2023) and to *multi-flow* multimodal models capable of supporting multiple generation flows within a single model (Hu et al., 2023; Xu et al., 2023; Le et al., 2025). Among various DGMs, energy-based models (EBMs) are a particularly flexible class of generative models and are known for being expressive in capturing contextual relationships in data space (Du et al., 2020; Gao et al., 2020; Cui et al., 2023a). Despite these strengths, most EBMs have primarily focused on single-modality settings and remain largely underexplored in the multimodal domain, falling behind other generative approaches.

Learning EBMs via maximum likelihood estimation (MLE) requires obtaining EBM samples, typically through Markov Chain Monte Carlo (MCMC) methods such as Langevin dynamics. However, noise-initialized Langevin dynamics is often ineffective, as it may take a long time to mix between different local modes. To mitigate this issue, prior *single-modal* EBMs (Xie et al., 2018; 2021; 2022; Cui & Han, 2023) have explored using complementary generator models to provide informative initializations for EBM sampling, thereby enabling more effective EBM learning. Different to single-modality, *multimodal* data additionally contain complex *inter-modal* dependencies (i.e., relationships shared across modalities) and modal-specific variations (i.e., inductive biases to each modality), making *informative* and *coherent* multimodal initializations particularly critical for multimodal EBM sampling and learning.

Another line of work, the shared latent variable generative model (a.k.a the shared generator model) (Wu & Goodman, 2018; Shi et al., 2019), has emerged as a promising approach for multimodal modeling. These models factorize a shared latent space along with modality-specific generation models that map the low-dimensional latent space to high-dimensional data space. The shared latent variable is learned to capture the shared representations across different modalities, while the generation models can preserve the unique characteristics of each modality. Learning the shared generator model via MLE typically requires MCMC posterior sampling, where noise-initialized Langevin dynamics is often hard to effectively traverse the shared latent space (Nijkamp et al., 2020). Alternatively, Multimodal VAEs (Palumbo et al., 2023; 2024) have developed variational learning schemes by introducing a joint inference model to approximate the generator posterior. However, both the shared generator and joint inference model are typically parameterized as unimodal Gaussian (or Laplace) distributions, which can be limited in expressivity for multimodal data and posterior distributions (Pang et al., 2021), leading to suboptimal models learned and degraded synthesis quality.

To address these limitations, we propose a novel learning scheme that can seamlessly integrate the multimodal EBM, the shared generator, and the joint inference model into a joint probabilistic framework. Specifically, the shared generator model is learned to match the EBM density, allowing it to provide informative and coherent initializations for MCMC EBM sampling. The joint inference model is learned to match the generator posterior, offering well-initialized starting points for MCMC posterior sampling. By jump-starting EBM and generator posterior sampling, the shared generator can be learned more effectively, which further facilitates EBM sampling and training. The resulting EBM samples and posterior samples in turn provide revision signals that continuously guide and refine both initializer models. This cooperative interplay among the three models yields effective sampling, accurate posterior inference, and stable training dynamics, leading to a multimodal EBM capable of realistic and coherent multimodal synthesis.

Our contributions can be summarized as: (**i**) We present a novel learning methodology that facilitates effective EBM sampling and learning toward multimodality. (**ii**) We integrate the multimodal EBM, shared generator, and joint inference model into a unified probabilistic framework, interleaving their MLE updates so that each component benefits from the others. (**iii**) We conduct extensive experiments, demonstrating superior performance of our multimodal EBM and effectiveness of our learning method.

## 2 PRELIMINARY

### 2.1 MULTIMODAL ENERGY-BASED MODEL

Let $\mathbf{X} = \{\mathbf{x}_1, \ldots, \mathbf{x}_M\}$ denote an observed multimodal data example consisting of $M$ modalities, and let $p_{\text{data}}(\mathbf{X})$ represent the unknown empricial data distribution. Energy-based models (EBMs) (Du et al., 2020; Gao et al., 2020; Cui et al., 2023b) represent a flexible class of generative models that define an undirected probability distribution

$$\pi_\alpha(\mathbf{X}) = \frac{1}{\mathbf{Z}(\alpha)} \exp\left[F_\alpha(\mathbf{X})\right] \tag{1}$$

where $\mathbf{Z}(\alpha) = \int_{\mathbf{X}} \exp\left[F_\alpha(\mathbf{X})\right] d\mathbf{X}$ is the intractable normalizing constant (or the partition function), and $F_\alpha(\mathbf{X})$ is the energy function parameterized by learnable parameters $\alpha$. For multimodal $\mathbf{X}$, the energy function takes all inputs $\{\mathbf{x}_1, \ldots, \mathbf{x}_M\}$ and outputs an energy value.

The EBMs offer considerable modeling flexibility; however, their application to multimodal data remains relatively underexplored. A key challenge lies in designing effective energy functions that can jointly capture the structure and dependencies across heterogeneous modalities. In this work, we adopt a simple yet general design for the energy function: $F_\alpha(\mathbf{X}) = \bar{f}([f_1(\mathbf{x}_1), \ldots, f_M(\mathbf{x}_M)])$, where each $f_i$ maps modality $\mathbf{x}_i$ into a fixed-dimensional feature vector, and $\bar{f}$ aggregates the concatenated features to produce the final energy score. While more sophisticated architectural choices may further enhance performance, our focus is on the learning methodology, and we leave architectural optimization for future work. Implementation details are provided in the Appendix. D.

**MLE Learning of $\pi_\alpha(\mathbf{X})$:** Given $N$ multimodal data $\{\mathbf{X}_1, \ldots, \mathbf{X}_N\}$ drawn from $p_{\text{data}}(\mathbf{X})$, the EBM can be trained via maximum likelihood estimation (MLE). The log-likelihood is computed as

$\mathcal{L}_\pi(\alpha) = \frac{1}{N}\sum_{i=1}^N \log \pi_\alpha(\mathbf{X}_i)$. When $N$ becomes sufficiently large, maximizing $\mathcal{L}_\pi(\alpha)$ is equivalent to minimizing the KL divergence between the true data density and EBM density, i.e.,

$$-\mathcal{L}_\pi(\alpha) = D_{\mathrm{KL}}(p_{\mathrm{data}}(\mathbf{X})||\pi_\alpha(\mathbf{X})) \tag{2}$$

$$\text{where} \quad \frac{\partial}{\partial\alpha}\mathcal{L}_\pi(\alpha) = \mathbb{E}_{p_{\mathrm{data}}(\mathbf{X})}\left[\frac{\partial}{\partial\alpha}F_\alpha(\mathbf{X})\right] - \mathbb{E}_{\pi_\alpha(\mathbf{X})}\left[\frac{\partial}{\partial\alpha}F_\alpha(\mathbf{X})\right]$$

**EBM Sampling.** Computing Eqn. 2 requires EBM samples, i.e., $\mathbf{X} \sim \pi_\alpha(\mathbf{X})$, which can be achieved via MCMC methods, such as Langevin dynamics (Neal et al., 2011) that iteratively updates

$$\mathbf{X}^{k+1} = \mathbf{X}^k + s\frac{\partial}{\partial\mathbf{X}^k}\log\pi_\alpha(\mathbf{X}^k) + \sqrt{2s}\cdot\epsilon^k \tag{3}$$

where $k$ denotes the iteration index, $s$ is the step size, and $\epsilon^k$ is Gaussian noise. In the limit as $s \to 0$ and $k \to \infty$, this process will converge to the stationary distribution $\pi_\alpha(\mathbf{X})$ (Neal et al., 2011).

Common practices typically adopt short-run Langevin dynamics (Nijkamp et al., 2019; Cui & Han, 2024), which performs certain steps (e.g., $k = 30$) of Langevin dynamics to generate approximate EBM samples. While this approach has been shown to yield meaningful learning signals, it remains challenging to draw effective EBM samples when starting from non-informative initializations[1] (Grathwohl et al., 2021; Kumar et al., 2019). More importantly, the structure of multimodal data introduces additional challenges. The input $\mathbf{X} = \{\mathbf{x}_1,\dots,\mathbf{x}_M\}$ involves complex inter-modal dependencies (i.e., relationship between modalities), and successful sampling should discover and mix among coherent local modes that reflect consistent relationships across modalities. As a result, effective initialization and sampling become particularly critical for multimodal data.

## 2.2 Multimodal Shared Latent Variable Generative Model

On the other hand, shared latent variable generative models (also referred to as shared generator models) have emerged as a promising approach for modelling complex multimodal data distributions (Wu & Goodman, 2018; Shi et al., 2019). Let $\mathbf{z}$ denote the low-dimensional latent variables. The shared generator model defines a joint distribution over multimodal inputs $\{\mathbf{x}_1,\dots,\mathbf{x}_M\}$ as

$$p_\omega(\mathbf{X},\mathbf{z}) = p_\omega(\mathbf{X}|\mathbf{z})p_0(\mathbf{z}) \quad \text{where}$$
$$p_\omega(\mathbf{X}|\mathbf{z}) = p_{\omega_1}(\mathbf{x}_1|\mathbf{z})p_{\omega_2}(\mathbf{x}_2|\mathbf{z})\cdots p_{\omega_M}(\mathbf{x}_M|\mathbf{z}) \tag{4}$$

Here, $p_0(\mathbf{z})$ is the prior distribution (e.g., Gaussian or Laplace distribution) over a shared latent variable $\mathbf{z}$, and $p_\omega(\mathbf{X}|\mathbf{z})$ is the conditional likelihood given such shared latent variable and factorizes a product of $M$ modality-specific generation models. Each $p_{\omega_i}(\mathbf{x}_i|\mathbf{z}) \sim \mathcal{N}(\mu_\omega(\mathbf{z}),\sigma^2 I_d)$ represents a conditional Gaussian parameterized by $\omega_i$, mapping the low-dimensional latent space to the high-dimensional data space for each modality.

This shared generator model is designed to capture modality-invariant representations (i.e., high-level semantics) across different modalities through the shared latent space $\mathbf{z}$, while also being capable of modelling modality-specific biases through separate generation models for each modality.

**Multimodal Joint Inference Model**. For learning Eqn. 4, multimodal VAEs (Sutter et al., 2020; Hwang et al., 2021; Palumbo et al., 2023; 2024) employ variational learning schemes by introducing a joint inference model. For multimodal data, factorizing effective joint inference models remains challenging and is an active research area (see details in Sec. 4). Among various methods, one major paradigm is the mixture-of-experts (MoE) (Shi et al., 2019) defined as

$$q_\phi(\mathbf{z}|\mathbf{X}) = \frac{1}{M}\sum_{i=1}^M q_{\phi_i}(\mathbf{z}|\mathbf{x}_i) \tag{5}$$

Each $q_{\phi_i}(\mathbf{z}|\mathbf{x}_i) \sim \mathcal{N}(\mu_{\phi_i}(\mathbf{x}_i),V_{\phi_i}(\mathbf{x}_i))$ is modeled as conditional Gaussian (or Laplace), where $\mu_{\phi_i}(\mathbf{x}_i)$ and $V_{\phi_i}(\mathbf{x}_i)$ denote the mean and diagonal covariance matrix parameterized by $\phi_i$.

This mixture-based joint inference model offers a tractable approximation to the generator posterior, particularly useful in scenarios with missing modalities. However, both the mixture formulation and the assumed individual posteriors are limited in statistical expressivity. They often induce an overly smooth latent space, which may fail to capture the intricate structure of the true multimodal generator posterior, ultimately resulting in a suboptimal generator model (see analysis in Sec. 3.1).

---

[1]i.e., $\mathbf{X}^{k=0}$ drawn from unit Gaussian or Uniform distribution.

## 3 METHODOLOGY

### 3.1 REVISITING LEARNING OF THE SHARED LATENT VARIABLE GENERATIVE MODEL

**From MLE Perspective:** Consider maximizing the log-likelihood of the shared generator model, i.e., $\mathcal{L}_p(\omega) = \frac{1}{N} \sum_{i=1}^{N} \log p_\omega(\mathbf{X}_i)$, where $p_\omega(\mathbf{X}_i) = \int_{\mathbf{z}} p_\omega(\mathbf{X}, \mathbf{z}) d\mathbf{z}$ is its marginal distribution. With a sufficiently large number of $N$, it is equivalent to minimizing the KL-divergence as

$$\mathcal{L}_p(\omega) = D_{\mathrm{KL}}(p_{\mathrm{data}}(\mathbf{X}) || p_\omega(\mathbf{X})) \tag{6}$$

$$\text{where} \quad \frac{\partial}{\partial \omega} \mathcal{L}_p(\omega) = \mathbb{E}_{p_{\mathrm{data}}(\mathbf{X}) p_\omega(\mathbf{z}|\mathbf{X})} \left[ \frac{\partial}{\partial \omega} \log p_\omega(\mathbf{X}, \mathbf{z}) \right]$$

Here, $p_\omega(\mathbf{z}|\mathbf{X})$ is the generator posterior. To shed further light, we can decompose it into

$$p_\omega(\mathbf{z}|\mathbf{X}) = \frac{p_\omega(\mathbf{X}|\mathbf{z}) p_0(\mathbf{z})}{p_\omega(\mathbf{X})} = \frac{p_0(\mathbf{z})}{p_\omega(\mathbf{X})} \prod_{i=1}^{M} \frac{p_{\omega_i}(\mathbf{z}|\mathbf{x}_i) p_{\omega_i}(\mathbf{x}_i)}{p_0(\mathbf{z})} \propto \frac{\prod_{i=1}^{M} p_{\omega_i}(\mathbf{z}|\mathbf{x}_i)}{\prod_{i=1}^{M-1} p_0(\mathbf{z})} \tag{7}$$

which reveals that the generator posterior is effectively a product of individual posteriors, modulated by the prior, leading to sharp and complex structures in the latent space. Approximating this product-based posterior using a mixture-based joint inference model (Eqn. 5) can be suboptimal due to the smoothing effect inherent in averaging (Daunhawer et al., 2021). Moreover, the unimodal $q_{\phi_i}(\mathbf{z}|\mathbf{x}_i)$ (Gaussian or Laplace distribution) can be limited in statistical expressivity and may fail to capture the intricate structure of the complex individual posteriors (Pang et al., 2021; Xie et al., 2022).

**MCMC Posterior Sampling.** Alternatively, one can directly obtain posterior samples by MCMC methods, such as Langevin dynamics (Han et al., 2017; Kong et al., 2024a;b), i.e.,

$$\mathbf{z}^{k+1} = \mathbf{z}^k + s \frac{\partial}{\partial \mathbf{z}^k} \log p_\omega(\mathbf{z}^k|\mathbf{X}) + \sqrt{2s} \cdot \epsilon^k \tag{8}$$

The target distribution is the generator posterior $p_\omega(\mathbf{z}|\mathbf{X})$, and the gradient term can be computed as $\frac{\partial}{\partial \mathbf{z}} \log p_\omega(\mathbf{z}|\mathbf{X}) \propto \frac{\partial}{\partial \mathbf{z}} \log p_\omega(\mathbf{X}|\mathbf{z}) p_0(\mathbf{z})$. The log-likelihood gradient decomposes as $\log p_\omega(\mathbf{X}|\mathbf{z}) = \sum_{i=1}^{M} \log p_{\omega_i}(\mathbf{x}_i|\mathbf{z})$, which updates shared latent variable $\mathbf{z}$ to explain all modality observations $\{\mathbf{x}_1, \ldots, \mathbf{x}_M\}$. As $s \to 0$ and $k \to \infty$, this process converges to the stationary distribution as generator posterior $p_\omega(\mathbf{z}|\mathbf{X})$ (Neal et al., 2011).

While MCMC-based posterior sampling can yield more accurate approximations than variational methods, it often suffers from poor mixing and slow convergence when using short-run Langevin dynamics (e.g., $k = 10$) initialized from non-informative points[2] (Nijkamp et al., 2020). More critically, the product-based formulation of the generator posterior requires complete observations from all modalities, and the individual unimodal posteriors are often undertrained and poorly calibrated. With inconsistent initializations, this becomes ill-defined in cross-modal inference scenarios, where only a subset of modalities is available (Shi et al., 2019; Daunhawer et al., 2021).

### 3.2 MULTIMODAL COOPERATIVE LEARNING VIA MCMC-REVISION

To address these limitations, we propose a novel multimodal learning framework that jointly learns the multimodal EBM, shared generator, and joint inference model through a cooperative mechanism. Specifically, our approach leverages the *complementary* strengths of each component: the inference model provides informative and coherent initializations for MCMC posterior sampling; the shared generator model facilitates consistent multimodal samples for EBM sampling; and the EBM offers critical revisions for both the generator and inference model. By interleaving their respective MLE updates and integrating MCMC-based revision, each model benefits from the others, leading to improved sampling efficiency and more effective multimodal modelling

#### 3.2.1 REVISION SIGNAL OF DUAL-MCMC KERNEL

Denote $\mathcal{M}_\alpha^{k_\mathbf{X}}(\cdot)$ for Markov transition kernel of $k_\mathbf{X}$ steps on EBM density (Eqn. 3), and denote $\mathcal{M}_\omega^{k_\mathbf{z}}(\cdot)$ for Markov transition kernel of $k_\mathbf{z}$ steps on generator posterior (Eqn. 8). We specify two

---

[2]e.g., $\mathbf{z}^{k=0}$ drawn from a unit Gaussian or uniform distribution.

joint densities over the MCMC process as

$$\Omega_{\omega,\alpha}(\mathbf{X}, \mathbf{z}) = \mathcal{M}_{\alpha}^{k_{\mathbf{x}}} \cdot p_{\omega}(\mathbf{X}|\mathbf{z})p_0(\mathbf{z}), \quad \Phi_{\omega,\phi}(\mathbf{X}, \mathbf{z}) = p_{\mathrm{data}}(\mathbf{X}) \cdot \mathcal{M}_{\omega}^{k_{\mathbf{z}}} \cdot q_{\phi}(\mathbf{z}|\mathbf{X}) \quad (9)$$

where $\Omega_{\omega,\alpha}(\mathbf{X}, \mathbf{z})$ takes the initialization from the shared generator model and performs MCMC transition on the EBM density, leading to a more general marginal distribution (i.e., $\mathcal{M}_{\alpha}^{k_{\mathbf{x}}} p_{\omega}(\mathbf{X}) = \int_{\bar{\mathbf{X}}} \int_{\mathbf{z}} \mathcal{M}_{\alpha}^{k_{\mathbf{x}}}(\bar{\mathbf{X}})p_{\omega}(\bar{\mathbf{X}}, \mathbf{z})d\mathbf{z}d\bar{\mathbf{X}})$ compared to the Gaussian generator model. $\Phi_{\omega,\phi}(\mathbf{X}, \mathbf{z})$ takes the initialization from the multimodal joint inference model and performs MCMC transition on the generator posterior, leading to a more accurate latent posterior distribution (i.e., $\mathcal{M}_{\omega}^{k_{\mathbf{z}}} q_{\phi}(\mathbf{z}|\mathbf{X}) = \int_{\bar{\mathbf{z}}} \mathcal{M}_{\omega}^{k_{\mathbf{z}}}(\bar{\mathbf{z}}|\mathbf{X})q_{\phi}(\bar{\mathbf{z}}|\mathbf{X})d\bar{\mathbf{z}})$ compared to the mixture-based joint inference model and their unimodal (Gaussian or Laplace) individual posteriors.

Similar MCMC-revised densities are also adopted in prior cooperative methods (Xie et al., 2018; 2022; 2021; Cui & Han, 2023), , but these approaches are limited to single-modal data. In contrast, our framework targets the multimodal data $\mathbf{X}$, capturing not only the modality-specific sample $\mathbf{x}_i$ but also their inter-modal dependencies. $\Omega_{\omega,\alpha}(\mathbf{X}, \mathbf{z})$ leverages a shared generator model to produce coherent and consistent multimodal initializations, thereby enhancing the efficiency of multimodal EBM sampling and learning. Meanwhile, $\Phi_{\omega,\phi}(\mathbf{X}, \mathbf{z})$ employs a mixture-based joint inference model as the amortizer sampler, hence provides more accurate posterior samples than the variational learning schemes (Palumbo et al., 2023; Shi et al., 2019).

**Learning Objectives with MCMC-revised Densities.** With such two MCMC-revised densities at the $t$-th optimization step, i.e., $\Omega_{\omega_t,\alpha_t}(\mathbf{X}, \mathbf{z})$ and $\Phi_{\omega_t,\phi_t}(\mathbf{X}, \mathbf{z})$, they act as intermediate targets to facilitate cooperative learning among the three models. Each model is learned by minimizing the KL-divergence between the revised densities and their respective densities.

(**i**) for multimodal EBM $\alpha$, the learning objective $\mathcal{L}_{\pi}(\alpha)$ is

$$-\mathcal{L}_{\pi}(\alpha) = D_{\mathrm{KL}}(\Phi_{\omega_t,\phi_t}(\mathbf{X}, \mathbf{z})||\pi_{\alpha}(\mathbf{X})q_{\phi}(\mathbf{z}|\mathbf{X})) - D_{\mathrm{KL}}(\Omega_{\omega_t,\alpha_t}(\mathbf{X}, \mathbf{z})||\pi_{\alpha}(\mathbf{X})q_{\phi}(\mathbf{z}|\mathbf{X})) \quad (10)$$

$$\text{where} \quad \frac{\partial}{\partial \alpha}\mathcal{L}_{\pi}(\alpha) = \mathbb{E}_{\Phi_{\omega_t,\phi_t}(\mathbf{X},\mathbf{z})}\left[\frac{\partial}{\partial \alpha}F_{\alpha}(\mathbf{X})\right] - \mathbb{E}_{\Omega_{\omega_t,\alpha_t}(\mathbf{X},\mathbf{z})}\left[\frac{\partial}{\partial \alpha}F_{\alpha}(\mathbf{X})\right]$$

Given the gradient, our multimodal EBM can be learned by stochastic gradient ascent (SGA).

(**ii**) For shared generator model $\omega$, the learning objective $\mathcal{L}_{p}(\omega)$ is

$$-\mathcal{L}_{p}(\omega) = D_{\mathrm{KL}}(\Phi_{\omega_t,\phi_t}(\mathbf{X}, \mathbf{z})||p_{\omega}(\mathbf{X}, \mathbf{z})) + D_{\mathrm{KL}}(\Omega_{\omega_t,\alpha_t}(\mathbf{X}, \mathbf{z})||p_{\omega}(\mathbf{X}, \mathbf{z})) \quad (11)$$

$$\text{where} \quad \frac{\partial}{\partial \omega}\mathcal{L}_{p}(\omega) = \mathbb{E}_{\Phi_{\omega_t,\phi_t}(\mathbf{X},\mathbf{z})}\left[\frac{\partial}{\partial \omega}\log p_{\omega}(\mathbf{X}, \mathbf{z})\right] - \mathbb{E}_{\Omega_{\omega_t,\alpha_t}(\mathbf{X},\mathbf{z})}\left[\frac{\partial}{\partial \omega}\log p_{\omega}(\mathbf{X}, \mathbf{z})\right]$$

With such a gradient, the shared generator model can be learned by SGA.

(**iii**) For multimodal joint inference model $\phi$, the learning objecitve $\mathcal{L}_{q}(\phi)$ is

$$-\mathcal{L}_{q}(\phi) = D_{\mathrm{KL}}(\Phi_{\omega_t,\phi_t}(\mathbf{X}, \mathbf{z})||p_{\mathrm{data}}(\mathbf{X})q_{\phi}(\mathbf{z}|\mathbf{X})) + D_{\mathrm{KL}}(\Omega_{\omega_t,\alpha_t}(\mathbf{X}, \mathbf{z})||\pi_{\alpha}(\mathbf{X})q_{\phi}(\mathbf{z}|\mathbf{X})) \quad (12)$$

$$\text{where} \quad \frac{\partial}{\partial \phi}\mathcal{L}_{q}(\phi) = \mathbb{E}_{\Phi_{\omega_t,\alpha_t}(\mathbf{X},\mathbf{z})}\left[\frac{\partial}{\partial \phi}\log q_{\phi}(\mathbf{z}|\mathbf{X})\right] + \mathbb{E}_{\Omega_{\omega_t,\alpha_t}(\mathbf{X},\mathbf{z})}\left[\frac{\partial}{\partial \phi}\log q_{\phi}(\mathbf{z}|\mathbf{X})\right]$$

$$= \mathbb{E}_{\Phi_{\omega_t,\alpha_t}(\mathbf{X},\mathbf{z})}\left[\frac{\partial}{\partial \phi}\mathrm{LSE}(\log q_{\phi_i}(\mathbf{z}|\mathbf{x}_i))\right] + \mathbb{E}_{\Omega_{\omega_t,\alpha_t}(\mathbf{X},\mathbf{z})}\left[\frac{\partial}{\partial \phi}\mathrm{LSE}(\log q_{\phi_i}(\mathbf{z}|\mathbf{x}_i))\right]$$

Here, $\mathrm{LSE}(\cdot)$ denotes the log-sum-exp operation, i.e., $\log \sum_{i=1}^{M} \exp(\cdot)$, corresponding to the mixture-based joint inference model across $M$ modalities. By computing such a gradient, the inference model can be learned by SGA.

We provide Pytorch pseudocode in the Appendix. F for implementation clarity.

### 3.2.2 HOW DO MCMC-REVISED KERNELS CONNECT THREE MODELS?

These MCMC-revised kernels serve as bridges between the three components and guide how they interact during cooperative learning. Consider the long-run behavior of $\mathcal{M}_{\alpha}^{k_{\mathbf{x}}}(\cdot)$ and $\mathcal{M}_{\omega}^{k_{\mathbf{z}}}(\cdot)$ kernel, the marginal distribution induced by the generator and inference models converge as $\mathcal{M}_{\alpha}^{k_{\mathbf{x}}} p_{\omega}(\mathbf{X}) \to \pi_{\alpha_t}(\mathbf{X})$ and $\mathcal{M}_{\omega}^{k_{\mathbf{z}}} q_{\phi}(\mathbf{z}|\mathbf{X}) \to p_{\omega_t}(\mathbf{z}|\mathbf{X})$, respectively. Learning the multimodal EBM by minimizing

Eqn. 10 therefore seeks to approximate the true data distribution $p_{\text{data}}(\mathbf{X})$, while simultaneously contrasting itself with its own previous state $\pi_{\alpha_t}(\mathbf{X})$. Particularly, Eqn. 10 amounts to

$$-\mathcal{L}_\pi(\alpha) \equiv \underbrace{D_{\text{KL}}(p_{\text{data}}(\mathbf{X})||\pi_\alpha(\mathbf{X}))}_{\text{match data density}} - \underbrace{D_{\text{KL}}(\pi_{\alpha_t}(\mathbf{X})||\pi_\alpha(\mathbf{X}))}_{\text{criticize itself}} \tag{13}$$

This formulation reflects a form of *self-adversarial* learning: the EBM not only seeks to match the data distribution but also acts as a critic of its previous estimate, encouraging continual refinement. Importantly, this surrogate objective provides a tractable EBM learning with the partition function term $\log \boldsymbol{Z}(\alpha)$ canceled out, making learning more stable and efficient.

Similarly, minimizing the generator objective in Eqn. 11 can be interpreted as:

$$-\mathcal{L}_p(\omega) \equiv \underbrace{D_{\text{KL}}(p_{\text{data}}(\mathbf{X})||p_\omega(\mathbf{X}))}_{\text{match data density}} + \underbrace{D_{\text{KL}}(\pi_{\alpha_t}(\mathbf{X})||p_\omega(\mathbf{X}))}_{\text{match EBM density}} + \tag{14}$$

$$\underbrace{\mathbb{E}_{p_{\text{data}}(\mathbf{X})}\left[D_{\text{KL}}(p_{\omega_t}(\mathbf{z}|\mathbf{X})||p_\omega(\mathbf{z}|\mathbf{X}))\right] + \mathbb{E}_{\pi_{\alpha_t}(\mathbf{X})}\left[D_{\text{KL}}(p_{\omega_t}(\mathbf{z}|\mathbf{X})||p_\omega(\mathbf{z}|\mathbf{X}))\right]}_{\text{additional surrogate KL perturbation terms}}$$

That is, the shared generator model is trained to align not only with the true data distribution but also with the EBM-induced density. The surrogate KL perturbation terms further contribute to more tractable optimization by forming an upper bound on the marginal likelihood, i.e., majorization principle Han et al. (2019), where the latent variable $\mathbf{z}$ is treated as part of the complete data inferred from the current optimization step. Unlike prior cooperative learning frameworks designed for single-modality data, our approach explicitly models the multimodal structure. The shared generator plays a pivotal role in synthesizing coherent multimodal samples that not only facilitate EBM learning but also preserve consistent cross-modal representations. This cooperative interplay across components promotes effective sampling, accurate posterior inference, and stable training dynamics in the multimodal setting.

Learning the joint inference model by minimizing Eqn. 12 matches the corresponding latent samples from multimodal real data $\mathbf{X} \sim p_{\text{data}}(\mathbf{X})$ and multimodal synthesis $\mathbf{X} \sim \pi_{\alpha_t}(\mathbf{X})$. Specifically, given the optimal $\mathcal{M}_\alpha^{k_{\mathbf{X}}}(\cdot)$ and $\mathcal{M}_\omega^{k_{\mathbf{z}}}(\cdot)$ kernel Eqn. 12 becomes

$$-\mathcal{L}_q(\phi) \equiv \underbrace{\mathbb{E}_{p_{\text{data}}(\mathbf{X})}\left[D_{\text{KL}}(p_{\omega_t}(\mathbf{z}|\mathbf{X})||q_\phi(\mathbf{z}|\mathbf{X}))\right]}_{\text{real latent sample inference}} + \underbrace{\mathbb{E}_{\pi_{\alpha_t}(\mathbf{X})}\left[D_{\text{KL}}(p_{\omega_t}(\mathbf{z}|\mathbf{X})||q_\phi(\mathbf{z}|\mathbf{X}))\right]}_{\text{synthesis latent sample inference}} \tag{15}$$

Both terms encourage the joint inference model to approximate the generator posterior $p_{\omega_t}(\mathbf{z}|\mathbf{X})$ more closely. Since samples from the generator posterior (as characterized in Eqn. 7) tend to be more accurate than the mixture-based initializations, this objective drives the inference model to produce informative latent initializations. These consistent refined initializations, in turn, enhance the effectiveness of MCMC-based posterior sampling by allowing it to better traverse the latent space and discover local modes, ultimately improving the learning of the shared generator model.

## 3.3 MODEL GENERALIZATION TO MODAL-SPECIFIC LATENT VARIABLE

Notably, prior works (Sutter et al., 2020; Palumbo et al., 2023) extend the shared latent generative model by introducing additional modality-specific latent variables $\mathbf{W} = \{\mathbf{w}_1, \ldots, \mathbf{w}_M\}$, i.e.,

$$p_\omega(\mathbf{X}, \mathbf{z}, \mathbf{W}) = p_0(\mathbf{z}) \prod_{i=1}^{M} p_{\omega_i}(\mathbf{x}_i|\mathbf{z}, \mathbf{w}_i) p_0(\mathbf{w}_i) \quad q_\phi(\mathbf{z}, \mathbf{W}|\mathbf{X}) = q_{\phi_{\mathbf{z}}}(\mathbf{z}|\mathbf{X}) \prod_{i=1}^{M} q_{\phi_{\mathbf{w}_i}}(\mathbf{w}_i|\mathbf{x}_i) \tag{16}$$

The modality-specific latent variable $\mathbf{W}$ is introduced to capture inductive biases unique to each modality, thereby enhancing the representational capacity of the latent space and improving robustness in cross-modal inference scenarios. We also extend our proposed learning method to accommodate this advanced variant. The proposed learning framework remains fully applicable, demonstrating the flexibility in supporting alternative multimodal generative models. Efficiently, our MCMC-revised kernels are performed over multimodal data or latent variables simultaneously, and the inclusion of $\mathbf{W}$ does not introduce additional inter-loop overhead.

## 4 RELATED WORK

**Energy-based model.** EBMs offer high modeling flexibility. While most learning strategies rely on MLE schemes (Nijkamp et al., 2019; Du & Mordatch, 2019; Du et al., 2020; Xiao et al., 2020), recent studies explore amortized sampling using a generator model (Han et al., 2019; Grathwohl et al., 2021; Kumar et al., 2019; Luo et al., 2024), in which Luo et al. (2024) proposes learning conditional EBM, and Han et al. (2019); Grathwohl et al. (2021); Kumar et al. (2019) proposed learning marginal EBM with complementary models. Such amortized-MCMC methods differ fundamentally from our MCMC-based method in the learning objective (see further discussion in App. B.1). Cooperative learning approaches instead propose using a generator to initialize MCMC chains Xie et al. (2018; 2021), enabling more efficient training. However, these methods have only focused on single-modal data. For multimodal settings, complex inter-modal dependencies pose additional challenges for effective EBM sampling and learning.

**Multimodal VAE.** MVAE (Wu & Goodman, 2018) introduced the Product-of-Experts (PoE) formulation, which combines unimodal posteriors into a single joint inference, enabling scalable and efficient training. In parallel, MMVAE (Shi et al., 2019) proposed a Mixture-of-Experts (MoE) approach to improve robustness with missing modalities. These two paradigms were later unified in MoPoE (Sutter et al., 2020), which formulated a Mixture-of-Products to capture richer combinations of modality subsets. MoPoE (Sutter et al., 2021) balances the tradeoff between flexibility and expressivity by combining PoE and MoE. Beyond the inference structure, other directions focused on improving the latent space itself. MVTCAE (Hwang et al., 2021) enforced cross-modal consistency via total correlation regularization. MMVAE+ (Palumbo et al., 2023) incorporated modality-specific priors to increase representational flexibility, and MVEBM (Yuan et al., 2024) replaced the Gaussian prior with an energy-based prior to better capture latent structures. CMVAE (Palumbo et al., 2024) introduced clustering objectives to enforce semantic structure in the latent space. Its variant, D-CMVAE, integrates DiffuseVAE (Pandey et al., 2022a), applying diffusion-based refinement to modality-specific outputs to significantly improve generation quality. Similarly, ScoreMVAE (Wesego & Rooshenas, 2023) applies score-based refinement at the latent level. However, these second-stage refinement schemes operate outside the core generator and therefore do not inherently improve the shared generative model; additional refinement is still required to fully enhance multimodal synthesis.

## 5 EXPERIMENT

In this section, we aim to answer the following questions: (1) Can our multimodal EBM generate realistic and coherent multimodal synthesis? (2) Do the complementary models align well with their MCMC-revised samples? and (3) How important are coherent initializers in the multimodal setting? Additional experiments and supplementary results are provided in the Appendix. A.

**Experiment Setting.** Following the protocols in our variational counterpart, we benchmark our method on PolyMNIST (Thomas M. Sutter, 2021) and Caltech-Birds (CUB) Image-Captions (Shi et al., 2019). PolyMNIST consists of five modalities, each representing the same digit class with varying backgrounds and styles. CUB involves two modalities (image-caption pair) that exhibit abstract shared semantics and rich modality-specific variation, considered a more challenging benchmark (Shi et al., 2019; Palumbo et al., 2023; 2024). To ensure fair comparison, we utilize the same generator and inference network structures as used in baselines (Palumbo et al., 2023; 2024).

### 5.1 MULTIMODAL DATA MODELLING

We first assess our proposed method in producing high-quality and coherent multimodal synthesis. Both the shared generator model and the multimodal EBM are trained to approximate the empirical data distribution. If the generator model is well-trained, it can serve as an informative initializer for EBM sampling, thereby facilitating more efficient sampling and enabling more effective EBM learning. In the meantime, the mixture-based joint inference model acts as an informative amortizer, providing accurate and consistent latent initializations for posterior sampling, leading to a better-trained generator, further enhancing EBM sampling and learning. Overall, this cooperative interplay contributes to improved synthesis quality and semantic coherence across modalities.

To quantitatively evaluate synthesis coherence, we follow standard protocols and utilize pre-trained classifiers[3]. These classifiers assess whether the generated samples correctly match the digit class across modalities, with higher accuracy indicating better coherence among modalities. We compare against several strong variational baselines on both unconditional and conditional synthesis. Note that CMVAE result on CUB is taken from its diffusion-based variant (denoted as Diff-CMVAE) that integrates diffusion models to largely improve generation quality. We denote Ours-**W** for using the adapted generator and inference model (Eqn. 16).

| Conditional CUB ‖ | MVAE | MVTCAE | mmJSD | MoPoE | MMVAE | MMVAE+ | MVEBM | Diff-CMVAE | Ours | Ours-W |
|---|---|---|---|---|---|---|---|---|---|
| FID (↓) ‖ | 172.21 | 208.43 | 262.80 | 265.55 | 232.20 | 164.94 | 136.16 | 28.00 | **25.98** | **24.32** |

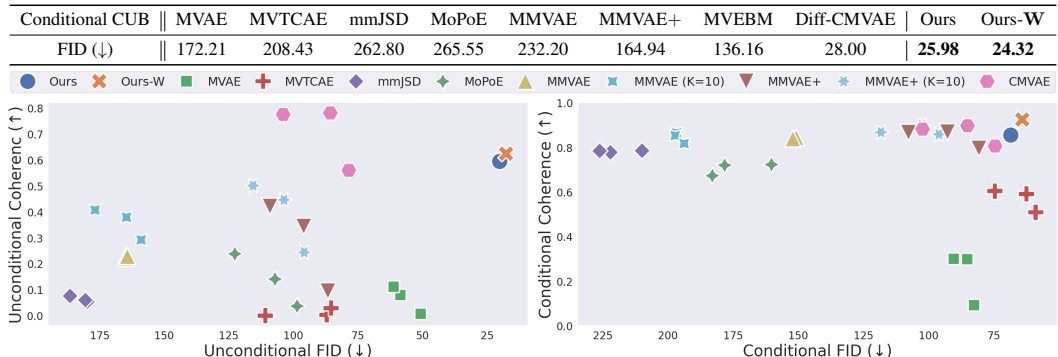

Figure 1: Comparison for unconditional and conditional multimodal synthesis on PolyMNIST (bottom), and comparison for conditional FID score on CUB (top).

To ensure a broad comparison over the landscape, Fig. 1 includes our variational baselines trained under different configurations (e.g., importance-weight sampling) used to optimize the performance. From Fig. 1, we observe that our method demonstrates superior performance for coherence and quality. Even compared to the diffusion Diff-CMVAE, our approach achieves lower FID scores while maintaining higher efficiency (versus diffusion sampling). These results validate the effectiveness of MCMC-revised cooperative learning in capturing shared semantics while preserving modality-specific details. Additional quantitative and qualitative results are provided in the Appendix. C.

## 5.2 ANALYSIS OF COMPLEMENTARY MODEL AND MCMC-REVISION

In our learning scheme, it is critical that the shared generator model and joint inference model closely match their corresponding MCMC-revised samples, ensuring that they can provide well-initialized states for EBM sampling and posterior sampling, respectively. In this section, we investigate whether these complementary models effectively align with their MCMC revisions. To do so, we visualize the trajectories of EBM sampling and posterior sampling, each initialized from the shared generator model and joint inference model, and refined through iterative Langevin dynamics.

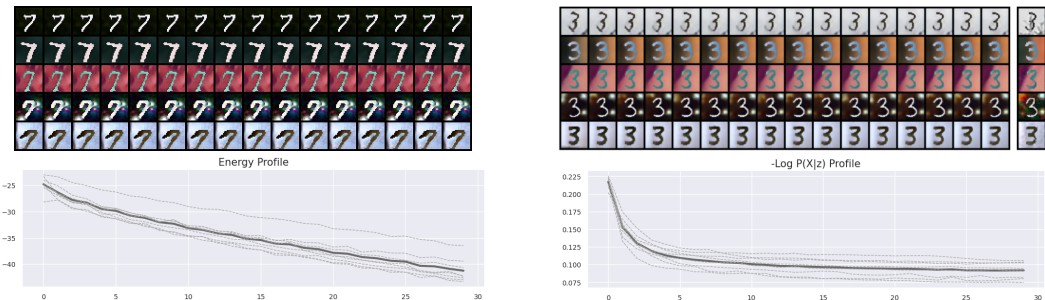

Figure 2: Trajectories of EBM sampling (**left**) and posterior sampling (**right**). Each row represents a different modality. The first column shows the initial states from their respective initializer models. We visualize every 2 steps with a total of 30 steps. The final column shows the outputs after MCMC refinement. The rightmost column in posterior sampling is the observed examples.

As shown in Fig. 2, the initializations are semantically coherent across modalities (representing the same digit class). As the Langevin dynamics progress, only minor refinements are observed,

[3]Pre-trained classifiers for each modality provided by Palumbo et al. (2024)

indicating that both the initializer models closely match their MCMC-revised samples. Nevertheless, the trending profile of the energy values $F_\alpha(\mathbf{X}^k)$ and the log-likelihood $\log p_\omega(\mathbf{X}|\mathbf{z}^k)$ continue to improve over Langevin iterations, highlighting the effectiveness of the MCMC-revised kernels in further refining and guiding the complementary models.

## 5.3 ANALYSIS OF SHARED GENERATOR MODEL

For multimodal data, our shared generator model (Eqn. 4) factorizes a single shared latent variable $\mathbf{z}$ to capture inter-modal dependencies, enabling coherent multimodal initializations for EBM sampling. To examine its importance, we replace the shared generator with $M$ *independent* generators $p_{\omega_i}(\mathbf{x}_i, \mathbf{z}_i)$, each with its own latent variable $\mathbf{z}_i$.

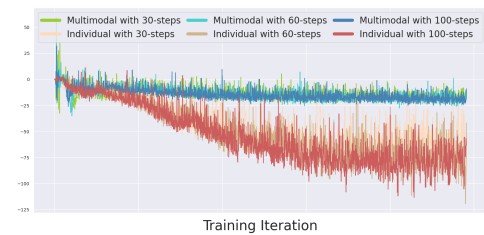

Training Iteration

We plot the resulting EBM loss (Eqn.10) profiles in Fig. 3. It can be seen that independent generators fail to produce coherent multimodal initializations, leading to fluctuating EBM loss and suboptimal learning. Even with more EBM sampling steps (e.g., $k_{\mathbf{X}} = 60$ and 100), the loss remains unstable. In contrast, the shared generator consistently yields stable learning dynamics, confirming its critical role in facilitating multimodal EBM learning.

Figure 3: EBM loss profile using the shared generator vs. independent generators.

## 5.4 ANALYSIS OF JOINT INFERENCE MODEL

Our joint inference model (Eqn. 5) serves as a multimodal latent initializer, producing consistent latent starting points for MCMC posterior sampling and improving the learning of the shared generator. To assess its impact, we replace it with $M$ *independent* inference models $p_{\phi_i}(\mathbf{z}_i|\mathbf{x}_i)$, each with its own latent variable $\mathbf{z}_i$.

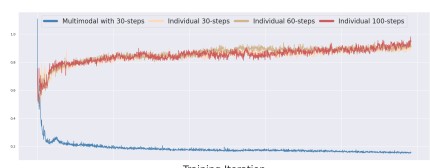

Training Iteration

Corresponding generator loss (Eqn. 11) profile is shown in Fig. 4. We observe that independent inference models fail to provide consistent multimodal latent initializations, resulting in higher generator loss that continues to increase even with more posterior sampling steps (e.g., $k_{\mathbf{z}} = 60$ and $k_{\mathbf{z}} = 100$). In contrast, our joint inference model achieves a steady decrease in generator loss, demonstrating its effectiveness in generator learning, which in turn enhances multimodal EBM learning.

Figure 4: Generator loss profiles for joint vs. independent inference models.

## 5.5 ABLATION STUDY

Table 1: MCMC Steps for FID and training Time.

| CUB | $k_{\mathbf{X}}$=60 | $k_{\mathbf{X}}$=10 | $k_{\mathbf{X}}$=30 and $k_{\mathbf{z}}$=30 | $k_{\mathbf{z}}$=10 | $k_{\mathbf{z}}$=60 |
|---|---|---|---|---|---|
| FID | 25.16 | 30.40 | **25.98** | 35.46 | 25.78 |
| Time (s / iteration) | 2.62 | 1.34 | 1.78 | 1.03 | 3.15 |

Table 2: FID and sampling Time.

| CUB | Generator | EBM ($k_{\mathbf{X}}$=30) |
|---|---|---|
| FID | 26.15 | **25.98** |
| Time (s / batch=100) | 0.001 | 0.08 |

**MCMC Steps of $\mathcal{M}_\alpha^{k_{\mathbf{X}}}$.** For EBM sampling, increasing $k_{\mathbf{X}}$ should benefit the Langevin dynamics to better explore the energy landscape and provide MCMC-revision signal for generator learning, leading to more effectively learned EBM.

**MCMC Steps of $\mathcal{M}_\omega^{k_{\mathbf{z}}}$.** Similarly, increasing the Langevin steps $k_{\mathbf{z}}$ of generator posterior sampling should render more accurate posterior samples to guide joint inference model, resulting in a more effectively learned shared generator, which in turn benefits multimodal EBM sampling and learning.

In Tab.1, increasing the Langevin steps from 10 to 30 yields a substantial improvement, while further increasing to 60 steps offers only marginal gains. We also report the FID and sampling cost of our shared generator model in Tab. 2, which shows higher generation quality at a much lower sampling cost compared to Diff-CMVAE (FID=28.00).

## 6 CONCLUSION

We propose a joint learning scheme that effectively learns the multimodal EBM by interweaving the MLE updates of the EBM, shared generator, and joint inference model through MCMC-based revision. The shared generator is learned to provide coherent initializations for MCMC EBM sampling, while the joint inference model is learned to offer starting points for MCMC posterior sampling. MCMC-revised samples, in turn, serve as revision signals, refining and guiding the shared generator and joint inference model, which facilitates effective multimodal EBM sampling and learning.

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

## A  ADDITIONAL EXPERIMENT

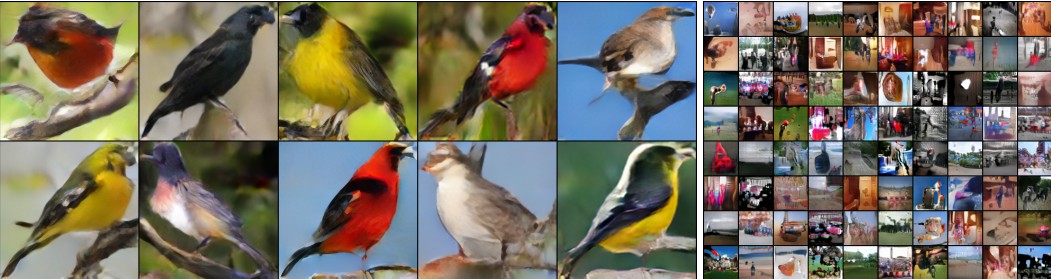

Figure 5: Unconditional synthesis on high-resolution CUB (left) and large-scale MSCOCO (right).

### A.1  SCALE-UP TO HIGH-RESOLUTION AND LARGE-SCALE DATASET

We test the scalability of our proposed method on the challenging high-resolution image (256x256) CUB data and the large-scale MSCOCO datasets. To better understand and assess the effectiveness endowed by our proposed learning method, we use the same network structures for all experiments. We visualize the unconditional and conditional multimodal synthesis in Fig. 5, suggesting our method effectively scales to higher resolutions and large-scale datasets while maintaining faithful multimodal synthesis quality. To further quantify this performance, we evaluate the generation quality and show results in Tab.3.

Table 3: FID on challenging dataset.

|  | Ours (Gen) | Ours (EBM) | MMVAE+ |
|---|---|---|---|
| CUB (256x256) | 56.32 | 55.81 | 213.74 |
| MSCOCO | 68.94 | 68.10 | 187.22 |

## A.2 LATENT SPACE INTERPOLATION

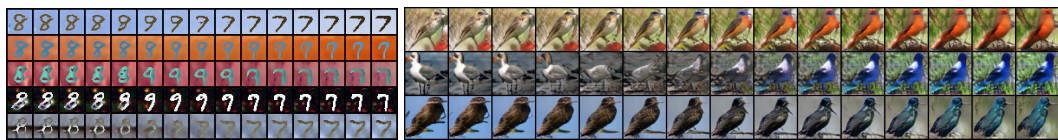

Figure 6: Visualization of unconditional synthesis via Latent space interpolation.

We evaluate whether the shared generator model can produce smooth interpolations in the shared latent space, leading to gradual transitions in the multimodal data space. To this end, we perform linear interpolation in the latent space, $\tilde{\mathbf{z}} = (1 - \alpha) \cdot \mathbf{z}_1 + \alpha \cdot \mathbf{z}_2$. As shown in Fig. 6, the shared generator produces smooth and coherent transitions across modalities, indicating its ability to capture shared semantics and effectively explore the energy landscape.

Table 4: Coherence with MCMC refinement.

| Number of Modality | 1 ($n = 0$) | 2 ($n = 1$) | 3 ($n = 2$) | 4 ($n = 3$) |
|---|---|---|---|---|
| Coherence | 0.921 | 0.930 | 0.938 | 0.940 |

Table 5: Accuracy for latent classifier.

| Method | Ours | MVAE | MMVAE | MoPoE |
|---|---|---|---|---|
| Accuracy | 0.962 | 0.926 | 0.835 | 0.944 |

## A.3 MCMC REFINEMENT ON CROSS-MODAL INFERENCE

Following the evaluation protocols of our variational baselines, we assessed conditional coherence when only one modality is available in Fig. 1 in the main text. In our framework, MCMC posterior sampling offers an additional capability: it can refine latent variables inferred from multiple subsets of available modalities. This refinement step, which iteratively adjusts the latent variables toward better cross-modal consistency, is, however, not feasible for standard variational approaches.

In particular, given $\{\mathbf{x}_i, \ldots, \mathbf{x}_{i+n}\}$ modalities available, we first obtain $\mathbf{z} \sim q_{\phi_j}(\mathbf{z}|\mathbf{x}_j)$ from arbitary one of them (i.e., $j \in \{i, \ldots, i+n\}$), and then we refine $\mathbf{z}$ with all $\{\mathbf{x}_i, \ldots, \mathbf{x}_{i+n}\}$ with Eqn. 8, so that we can generate the missing modalities with better coherence. We report our results in Tab. 4, where the conditional coherence becomes better with increasing number of available modalities.

## A.4 LATENT CLASSIFICATION

We further examine whether the inferred latent variables capture shared high-level semantics across modalities. Following (Sutter et al., 2021), we train latent classifiers on the inferred latent variables and measure classification accuracy. If the latent space effectively encodes shared semantic information, these classifiers should achieve high accuracy. Using our mixture-based joint inference model, we report the classification accuracy averaged over all modalities in Tab. 5.

## B THEORECTICAL DERIVATION

For MLE learning of the EBM objective (Eqn. 2), the gradient is derived as

$$
\begin{aligned}
\frac{\partial}{\partial \alpha} \mathcal{L}_\pi(\alpha) &= \mathbb{E}_{p_{\text{data}}(\mathbf{X})} \left[ \frac{\partial}{\partial \alpha} \log \pi_\alpha(\mathbf{X}) \right] \\
&= \mathbb{E}_{p_{\text{data}}(\mathbf{X})} \left[ \frac{\partial}{\partial \alpha} F_\alpha(\mathbf{X}) \right] - \frac{\partial}{\partial \alpha} \log \mathbf{Z}(\alpha)
\end{aligned}
$$
(17)

where $\frac{\partial}{\partial \alpha} \log \boldsymbol{Z}(\alpha)$ is derived as

$$
\begin{aligned}
\frac{\partial}{\partial \alpha} \log \boldsymbol{Z}(\alpha) &= \frac{1}{\boldsymbol{Z}(\alpha)} \int \frac{\partial}{\partial \alpha} \exp\left[F_\alpha(\mathbf{X})\right] d\mathbf{X} \\
&= \int \pi_\alpha(\mathbf{X}) \frac{\partial}{\partial \alpha}\left[F_\alpha(\mathbf{X})\right] d\mathbf{X} \qquad (18) \\
&= \mathbb{E}_{\pi_\alpha(\mathbf{X})}\left[\frac{\partial}{\partial \alpha} F_\alpha(\mathbf{X})\right]
\end{aligned}
$$

By applying Eqn. 18 to Eqn. 17, we have derived Eqn. 2.

For MLE learning of the shared generator objective (Eqn. 6), the gradient is derived as

$$
\begin{aligned}
\frac{\partial}{\partial \omega} \log p_\omega(\mathbf{X}) &= \mathbb{E}_{p_\omega(\mathbf{z}|\mathbf{X})}[\frac{\partial}{\partial \omega} \log p_\omega(\mathbf{X})] \\
&= \mathbb{E}_{p_\omega(\mathbf{z}|\mathbf{X})}[\frac{\partial}{\partial \omega} \log p_\omega(\mathbf{X})] + \mathbb{E}_{p_\omega(\mathbf{z}|\mathbf{X})}[\frac{\partial}{\partial \omega} \log p_\omega(\mathbf{z}|\mathbf{X})] \qquad (19) \\
&= \mathbb{E}_{p_\omega(\mathbf{z}|\mathbf{X})}[\frac{\partial}{\partial \omega} \log p_\omega(\mathbf{X}, \mathbf{z})]
\end{aligned}
$$

where $\mathbb{E}_{p_\omega(\mathbf{z}|\mathbf{X})}[\frac{\partial}{\partial \omega} \log p_\omega(\mathbf{z}|\mathbf{X})] = \int p_\omega(\mathbf{z}|\mathbf{X})[\frac{\partial}{\partial \omega} \log p_\omega(\mathbf{z}|\mathbf{X})]d\mathbf{z} = \frac{\partial}{\partial \omega} \int p_\omega(\mathbf{z}|\mathbf{X})d\mathbf{z} = 0$.

### B.1 COMPARED TO AMORIZED-MCMC METHOD

Several recent advances have investigated EBM learning without explicit MCMC sampling Grathwohl et al. (2021); Han et al. (2019); Luo et al. (2024); Schröder et al. (2023). These works study *single-modal* EBMs that employ amortized samplers to replace MCMC, thereby avoiding iterative sampling. In contrast, our focus is on the *multimodal setting*, which introduces two additional challenges: (i) effectively capturing the shared inter-modal relationships across heterogeneous modalities, and (ii) mitigating the mismatch induced by multimodal joint inference models. To address these challenges, we incorporate MCMC revision as a key component of our cooperative framework, which allows both the EBM and the generator posterior to be iteratively refined by each other, ensuring coherent multimodal alignment that cannot be achieved by amortized single-pass updates.

Moreover, the inclusion of MCMC revision makes our **learning objectives fundamentally different** from previous amortizing formulations. For clarity, and to directly illustrate the difference in learning dynamics independent of modality notation, we denote $\Omega, \Phi$ as shorthand for MCMC-revised densities $\Omega_{\omega,\alpha}(\mathbf{X}, \mathbf{z}), \Phi_{\omega,\phi}(\mathbf{X}, \mathbf{z})$ and denote $Q = q_\phi(\mathbf{z}|\mathbf{X})p_{data}(\mathbf{X})$, $\Pi = \pi_\alpha(\mathbf{X})q_\phi(\mathbf{z}|\mathbf{X})$, $P = p_\omega(\mathbf{X}, \mathbf{z})$ for joint densities of amortized models. We denote AM for methods using amortized models without MCMC.

**Learning the EBM ($\alpha$):** The corresponding KL terms in our method (Eqn. 10) and AM are: $\min_\alpha KL(\Phi\|\Pi) - KL(\Omega\|\Pi)$ v.s. $\min_\alpha KL(Q\|\Pi) - KL(P\|\Pi)$. Our formulation leverages MCMC-revised samples (i.e., joint densities of $\Omega$ and $\Phi$), whereas AM relies solely on ancestral samples ($Q$ and $P$). Because our samples are refined by the EBM itself, they provide a more accurate approximation of the target energy landscape $KL(M_{\alpha_t}p_{\omega_t}(\mathbf{X})\|\pi_{\alpha_t}(\mathbf{X})) \leq KL(p_{\omega_t}(\mathbf{X})\|\pi_{\alpha_t}(\mathbf{X}))$. This results in more effective and stable EBM learning and leverages the contextual modelling capability of EBM to effectively guide the multimodal generator model.

**Learning the (shared) generator model ($\omega$):** For learning the generator model (Eqn. 11), KL terms for ours and AM are $\min_\theta KL(\Phi\|P) + KL(\Omega\|P)$ v.s. $\min_\theta KL(Q\|P) + KL(P\|\Pi)$. The learning dynamics differ substantially. In our case, the generator is trained with MCMC-revised latent samples, yielding a closer match to the true generator posterior: $KL(M_{\omega_t}q_{\phi_t}(\mathbf{z}|\mathbf{X})\|p_{\omega_t}(\mathbf{z}|\mathbf{X})) \leq KL(q_{\phi_t}(\mathbf{z}|\mathbf{X})\|p_{\theta_t}(\mathbf{z}|\mathbf{X}))$, which aims to address the mismatch between the generaetor posterior and joint inference model (analysis in Sec. 3.1). In addition, $KL(\Omega\|P)$ learns to match the revised MCMC samples from EBM-refined samples, while the Amortizer method intends to chase the major modes of $\pi_\alpha(\mathbf{X})$ through variational approximation (i.e., $KL(P\|\Pi)$). Hence, our generator directly learns from revised multimodal samples that can better capture inter-modal consistency.

**Learning the (joint) inference model ($\phi$):** For the inference model (Eqn. 12), learning objectives for ours and AM are: $\min_\phi KL(\Phi\|Q) + KL(\Omega\|\Pi)$ v.s. $\min_\phi KL(Q\|P) + KL(P\|\Pi)$. The two

approaches differ in both learning source and optimization target. Our inference network amortizes latent MCMC refinement on observed data (i.e., $KL(\Phi\|Q)ComplilerError$), while AM performs pure variational inference (i.e., $KL(Q|P)$). On generated samples, our model matches EBM-revised generator samples (i.e., $KL(\Omega\|\Pi)$), whereas AM directly uses ancestral generator outputs ($KL(P|\Pi)$), which can lead to sub-optimal inference quality.

# C  SUPPLEMENTARY RESULT

Corresponding to our Fig. 1, we additionally report quantitative results in Tab.6 and Tab.7, where we report only the best performance of our baselines. Qualitative results can be seen in Fig. 7, Fig. 8, Fig. 9, and Fig. 10, .

Table 6: Comparison of synthesis coherence. Table 7: Comparison of multimodal synthesis quality.

| Methods | PolyMNIST Unconditional | Conditional |
|---|---|---|
| MVAE | 0.112 | 0.301 |
| MVTCAE | 0.029 | 0.604 |
| mmJSD | 0.076 | 0.785 |
| MoPoE | 0.238 | 0.723 |
| MMVAE | 0.232 | 0.844 |
| MMVAE+ | 0.421 | 0.869 |
| MVEBM | 0.735 | 0.857 |
| CMVAE | 0.781 | 0.897 |
| **Ours** | **0.594** | **0.855** |
| **Ours-W** | **0.624** | **0.921** |

| Methods | PolyMNIST Unconditional | Conditional |
|---|---|---|
| MVAE | 50.65 | 82.59 |
| MVTCAE | 85.43 | 58.95 |
| mmJSD | 179.76 | 178.27 |
| MoPoE | 98.56 | 160.29 |
| MMVAE | 164.29 | 150.83 |
| MMVAE+ | 86.64 | 80.75 |
| MVEBM | 75.43 | 70.45 |
| CMVAE | 78.52 | 74.53 |
| **Ours** | **20.12** | **68.52** |
| **Ours-W** | **17.65** | **64.12** |

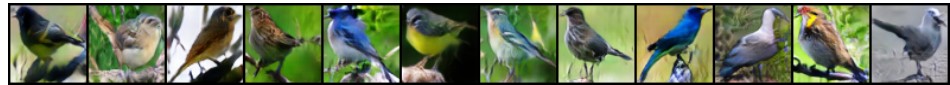

1. A black bird is up with a short, short bill.
2. The bird has a small surface and ooak tree which are black yellowed branches.
3. This bird has yellow with brown on its chest and has a very short beak.
4. This bird has wings that are black and have a brown crown.
5. This is a blue bird bird with white chest.
6. The bird has a green chest and black eye rings.
7. This particular bird has a belly that has white and yellow color.
8. The bird has a small brown bill with brown shoulder that also appear to be juvenile.
9. A blue bird with a chevron and something.
10. This bird has a white neck and wings that are grey and has a short bill.
11. This bird is brown coloured with a redhead and has a long crest.
12. This bird is white and grey in color, with it having few black wings.

Figure 7: Unconditional generation on CUB.

Input: this bird is shiny black, and blue in color, with a black beak.

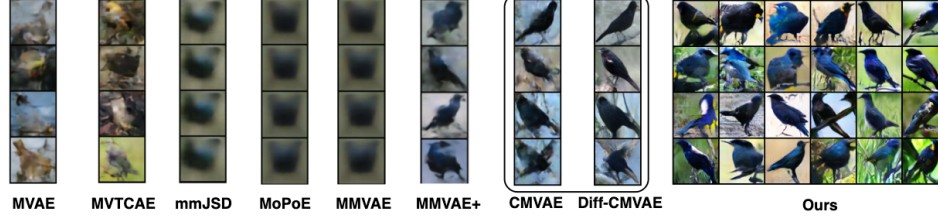

MVAE    MVTCAE    mmJSD    MoPoE    MMVAE    MMVAE+    CMVAE    Diff-CMVAE    Ours

Figure 8: Conditional generation on CUB. Baseline results are taken from (Palumbo et al., 2023). CMVAE and Diff-CMVAE results are reproduced with codes provided by Palumbo et al. (2024); Pandey et al. (2022b).

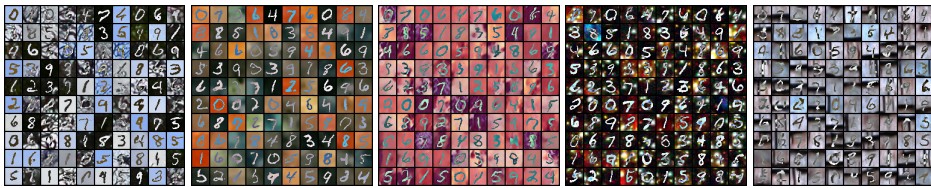

Figure 9: Unconditional generation on PolyMNIST.

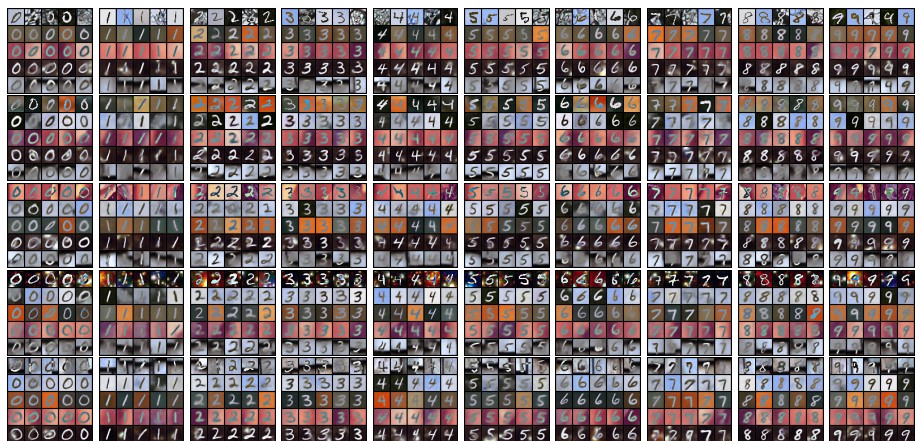

Figure 10: Conditional generation on PolyMNIST. From top to bottom, available modality from 1 to 5. In each block, the first row shows the given input modality, while the subsequent rows display the generated outputs for the remaining missing modalities.

# D  IMPLEMENTATION

| EBM Block (in_ch, out_ch, downsample, head) |
| --- |
| Input: x |
| ReLU if head |
| Conv(in_ch, out_ch), ReLU, Conv(out_ch, out_ch) |
| Downsample(factor=2) if downsample |
| output: h |
| Input: x |
| Downsample(factor=2) if head |
| Conv(in_ch, out_ch) if downsample |
| Downsample(factor=2) if downsample and not head |
| output: y |
| output: h + y |

| EBM Network on PolyMNIST (nef) |
| --- |
| Input: **X** |
| h = concat(Conv(each **x**)) along channel dim |
| EBM Block(nc, nef, downsample=True, head=True) |
| EBM Block (nef, nef, downsample=True) |
| EBM Block (nef, nef, downsample=False) |
| EBM Block (nef, nef, downsample=False) |
| ReLU, Downsample(factor=8), Linear(nef, 1) |
| output: h |
| **EBM Network on CUB (nef)** |
| Input: **X** |
| img = Conv(img), txt=Linear(ReLu(Linear(txt emb))) |
| EBM Block(nc, nef, downsample=True, head=True) |
| EBM Block (nef, nef, downsample=True) |
| EBM Block (nef, nef, downsample=False) |
| EBM Block (nef, nef, downsample=False) |
| ReLU, Downsample(factor=8), Linear(Concat(h,txt), 1) |
| output: h |

Table 8: We use generator and inference network structures from (Palumbo et al., 2023; 2024). For our EBM energy function structures, we denote the operation of convolution as **Conv** (input channel, output channel, k=3, s=1, p=1), where k is the kernel size, s is the stride number, and p is padding value. We conduct Upsample and Downsample via *interpolate* and *avg_pool2d* operations.

## D.1  INFERENCE MECHISIM UNDER MISSING MODALITIES

Our framework follows the standard inference mechanism established in multimodal VAEs for handling missing modalities. Given any available modality $x_i$, we first infer its shared latent variable, and then use this latent variable to generate the missing modalities through the shared generator for $x_j$ where $j \neq i$. This mechanism is identical to prior multimodal VAE baselines, ensuring fair comparison and consistent inference behavior. In our experiments, we evaluate using the same infer-

ence mechanism as in baseline models to ensure fairness, and the results consistently show superior reconstruction and coherence.

## E  DISCLOSURE OF LLM INVOLVEMENT

The LLM was employed solely for limited grammar refinement. It was not used for content generation, analysis, methodological development, nor for any other contribution to this work.

## F  PYTORCH PSEUDOCODE

```
1  import torch as t
2  import torch.nn as nn
3
4  data_loader = get_dataloader(dataset, batch_size)
5  netG, netI, netE = get_networks(dataset)
6
7  optG = t.optim.Adam(netG.parameters(), lr=1e-3)
8  optE = t.optim.Adam(netE.parameters(), lr=4e-4)
9  optI = t.optim.Adam(netI.parameters(), lr=1e-3)
10
11  e_l_steps, e_l_step_size, e_n_step_size = 30, 0.1, 0.001
12  z_l_steps, z_l_step_size, z_n_step_size = 30, 0.1, 0.1
13
14  latent_dim = 32
15  pz = get_distribution(t.distributions.Normal, latent_dim)
16  qz = get_distribution(t.distributions.Normal, latent_dim)
17
18  dataset = "PolyMNIST"
19  batch_size = 256
20
21  mse = nn.MSELoss(reduction='none').cuda()
22
23  def log_mean_exp(value, dim=0, keepdim=False):
24      return t.logsumexp(value, dim, keepdim=keepdim) - math.log(value.size(dim))
25
26  def langevin_x(x_init):
27      x = [x.clone().detach().requires_grad(True) for x in x_init]
28
29      for steps in range(e_l_steps):
30          energy = netE(x)
31          energy = energy.sum()
32          grad = t.autograd.grad(energy, x)
33          for d, x_i in enumerate(x):
34              x_i.data = x_i.data - 0.5 * e_l_step_size * e_l_step_size * grad[d] +
                      e_n_step_size * t.randn_like(x_i).data
35
36      return [x_i.detach() for x_i in x]
37
38  def langevin_z(z_init, x_data):
39      z = [z.clone().detach().requires_grad(True) for z in z_init]
40      views = len(z_init)
41      for steps in range(z_l_steps):
42          recon_value = [[None for _ in range(views)] for _ in range(views)]
43
44          for e in range(views):
45              for d in range(views):
46                  rec = netG(z[e], v_idx=d)
47                  recon_value[e][d] = mse(rec, x_data[d])
48
49          nls = []
50          for r in range(views):
51              lpz = pz.log_prob(z[r])
```

```python
            nlpx = [px_u for px_u in recon_value[r]]
            nlpxu = t.stack(nlpxu).sum(0)
            nl = nlpxu - lpz
            nls.append(nlw)
        nls = t.stack(nls).mean(0)
        nls = nls.sum(0)

        grad = t.autograd.grad(nls, z)
        for d, z_i in enumerate(z):
            z_i.data = z_i.data - 0.5 * z_l_step_size * z_l_step_size * grad[d] +
                z_n_step_size * t.randn_like(z_i).data

    return [z_i.detach() for z_i in z]

for i, x in enumerate(data_loader):
    x = [x_i.cuda() for x_i in x]
    views = len(x)

    z_prior = pz.rsample()
    samples_init = netG(z_prior)
    samples_corr = langevin_x(samples_init)

    z_q_mu_init, z_q_lv_init = netI(x)
    z_q_init = qz(z_q_mu_init, z_q_lv_init)
    z_q_corr = langevin_z(z_q_init, x)

    optG.zero_grad()
    recon_value = [[None for _ in range(views)] for _ in range(views)]

    for e in range(views):
        for d in range(views):
            rec = netG(z[e], v_idx=d)
            recon_value[e][d] = mse(rec, x[d])

    nls = []
    for r in range(views):
        nlpx = [px_u for px_u in recon_value[r]]
        nlpxu = t.stack(nlpxu).sum(0)
        nls.append(nlpxu)
    nls = t.stack(nls).mean(0)
    nls = nls.mean(0)

    errS = mse(samples_init, samples_corr)
    errG = nls + errS
    errG.backward()
    optG.step()

    optI.zero_grad()
    z_p_mu, z_p_lv = netI(samples_corr)

    nlqz_true = []
    nlqz_fake = []
    for r in range(views):
        lqz_true = log_mean_exp(t.stack([sum_flat(qz_x.log_prob(z_q_corr[r])) for qz_x in
            qz(z_q_mu_init, z_q_lv_init)]))
        nlqz_true.append(- lqz_true)
        lqz_gen = log_mean_exp(t.stack([sum_flat(qz_x.log_prob(z_prior)) for qz_x in qz(
            z_p_mu, z_p_lv)]))
        nlqz_fake.append(- lqz_gen)

    nlqz_true = t.stack(nlqz_true).mean(0)
    nlqz_fake = t.stack(nlqz_fake).mean(0)
    errI = nlqz_true + nlqz_fake
    errI.backward()
    optI.step()
```

```
114
115    optE.zero_grad()
116    E_t = netE(x)
117    E_f = netE(samples_corr)
118    errE = (E_t - E_f) / (e_n_step_size/e_l_step_size)**2
119    errE.backward()
120    optE.step()
```

Listing 1: PyTorch code used in our experiments.

