# OpenReview forum: "Cooperative Multimodal Energy-based Model with MCMC Revision"
_ICLR.cc/2026/Conference — Submitted to ICLR 2026_

### Official Review · Reviewer_qamK · 2025-10-29

**Soundness:** 3
**Presentation:** 2
**Contribution:** 2
**Rating:** 4
**Confidence:** 5

**Summary:**

This paper proposes a method for training Energy-Based Models (EBMs) on multimodal data with a cooperative learning approach. It tries to solve the common problems in multimodal VAEs that arise from poor-quality distribution matching and low-quality sampling. The authors introduce a learning scheme where three models are trained together in a cooperative way. This helps the model generate high quality samples from good initialization points for the EBMs. The final multimodal EBM is capable of generating higher-quality and more coherent modality outputs on datasets such as PolyMNIST and CUB.

**Strengths:**

The paper tries to address some of the main problems in multimodal VAEs, which are:
1. Low-quality samples
2. Coherent conditional and unconditional generation

In addition, the paper also implements their approach in larger scale dataset provided in the appendix where multimodal VAEs can't scale. The paper demonstrates that a cooperative approach in the EBMs helps the model achieve a superior generative capability. Knowing that Energy-Based-Models are difficult to train with multiple hyperparameters to tune, the authors did a good job of getting decent results using multiple EBMs in a cooperative approach.

**Weaknesses:**

Some of the weaknesses of the paper include:

1. $\textbf{Clarity of Method}$

The paper doesn't explain well the training of the EBMs and how they fuse it with multimodal VAE training. They also don't mention how they perform inference during missing modalities. In general, the paper could be written more effectively to address the core methods, while moving some mathematical repetitions to the appendix.


2. $\textbf{Missing baselines}$

The paper compares well with current multimodal VAEs but since misses some related works that should be added as a baseline. [1] introduces using score-based multimodal autoencoders for multimodal VAEs, but it's not cited and compared as a baseline. Realizing the connection between score-based and energy-based models, and the use of additional EBM in that work, the baseline is necessary here. In addition, how does the proposed model perform when one of the EBM models is missing or initialized directly without Langevin sampling?

3. $\textbf{Limitations of EBMs}$

Energy-Based-Models are difficult to train and unstable. Additionally, they require sampling during both training and inference. These makes them generally undesirable. And in these work, multiple EBMs are used which increases that effect.


[1] Wesego, Daniel, and Amirmohammad Rooshenas. "Score-based multimodal autoencoders." arXiv preprint arXiv:2305.15708 (2023).

**Questions:**

Please refer to the Weakness section.

---

> ### Author Response · Authors · 2025-11-14
>
> We sincerely thank the reviewer for the thoughtful and constructive feedback. We appreciate the recognition of our work motivation and the helpful suggestions for clarification and improvement. Most general concerns are addressed in the **general responses**, and we provide specific clarifications below.
>
> ### Clarification on EBM modelling and training.
>
> In our proposed method, we consider a single multimodal EBM (Eqn. 1), where the energy function serves as a scalar function and takes all multimodal data as input. To address the inherent challenges in EBM learning, we incorporate a shared generator and a joint inference model as complementary initializer networks. These components provide informative and coherent initializations for multimodal EBM MCMC sampling, while the EBM supplies refinement feedback to improve both initializer models, forming a stable cooperative learning cycle (see more analysis in **general response G.2**). As illustrated in Figure 3 (Sec. 5.3), this design leads to stable EBM training, and we did not observe mode collapse or divergence during training.
>
> ### Comparison with Score-Based Multimodal Autoencoder [1]
>
> We thank the reviewer for pointing out this related work. We will include and discuss it explicitly in the revision. The Score-based Multimodal Autoencoder also introduces an additional energy term, but it applies score-matching training in the latent space during a second-stage optimization. This approach faces similar limitations to Diff-CMVAE (see **general response G.3**): the generator itself is not effectively learned and depends on a separate refinement model for high-quality synthesis, resulting in higher sampling cost and reduced efficiency. In contrast, our method instead focus on a joint learning scheme that unifies the multimodal EBM, shared generator, and joint inference model, where each model can be learned more effectively, leading to high-quality multimodal synthesis for both the EBM and shared generator model.

---

> > ### Comment · Reviewer_qamK · 2025-11-25
> > **Follow up**
> >
> > Dear Authors,
> >
> > Thank you for the clarification. I believe I should have said multiple MCMC sampling, which is not ideal. I also don't see the discussion and comparison with the score-based multimodal autoencoder in your PDF right now, as you have stated to include it. I also believe recent works on EBM have better approaches than the one you used, as stated in [1]. What can you say about those works which do not require sampling at test time and are more efficient?
> >
> > [1] Luo, Yihong, Siya Qiu, Xingjian Tao, Yujun Cai, and Jing Tang. "Energy-calibrated vae with test time free lunch." In European Conference on Computer Vision, pp. 326-344. Cham: Springer Nature Switzerland, 2024.

---

> > > ### Author Response · Authors · 2025-11-25
> > >
> > > We hope that the additional clarifications and comparisons provided above help address the remaining concerns. Our goal in this work is to develop a **principled and unified multimodal learning framework** that integrates EBM, shared generator models, and joint inference models in a cooperative manner. We believe that the proposed ideas offer a meaningful step toward efficient, coherent, and scalable multimodal generative modelling, and may serve as a useful foundation for future research in this area. We appreciate reviewer thoughtful engagement with our work, and we humbly ask for a reconsideration of the review score. We would be very happy to provide any further clarification or discussion if needed.

---

> ### Author Response · Authors · 2025-11-25
>
> We sincerely thank the reviewer for the follow-up and for the helpful clarifications. Please see our updated manuscript for our previous discussion regarding to (i) Comparison with No MCMC method (Appendix. B.1) (ii) add discussion of socre-based multimodal VAE (Sec. 4) .
>
> We provide further clarification for your concerns below and will incorporate the corresponding revisions into the updated manuscript for our discussion.
>
> ### MCMC Sampling
>
> We agree that MCMC sampling introduces computational overhead due to its iterative nature. While in the context of multimodality, MCMC revision plays a crucial role in our framework due to the mismatch between the variational posterior and generator posterior (see also **general response G.1, G.2**). To ensure efficiency, both the EBM and generator posterior employ *short-run MCMC chains* that are initialized from informative states provided by the generator and inference models. This design keeps the runtime overhead minimal while still leveraging the benefits of revision-based refinement. As reported in **Tab A**, our total training and sampling time remains *shorter than Diff-CMVAE*.
>
> ### Discussion with the Score-based Multimodal Autoencoder and Energy-calibrated VAE with Test-Time Free Lunch [1]
>
> The **Score-Based Multimodal Autoencoder** also introduces an energy-related term but performs score-matching in the latent space during a **second-stage optimization**, similar to the training pipeline of Diff-CMVAE. This design relies on an additional score-based refinement for synthesis refinement. Our framework instead performs **joint learning** of the multimodal EBM, shared generator, and joint inference model within a unified probabilistic framework, so that each component (including the generator) is optimized during cooperative training.
>
> We thank the reviewer for highlighting this recent work. The **Energy-Calibrated VAE (ECVAE)** [1] introduces a conditional EBM that calibrates the complementary VAE without MCMC sampling at test time. We see this direction as highly complementary to ours. While ECVAE focuses on a single-modality setting and learns conditional EBM, targeting local correction of mode around the amortized samples, our framework addresses the multimodal learning problem by formulating marginal multimodal EBM, where maintaining *cross-modal coherence* requires iterative interaction between modalities. Importantly, in our joint framework, the shared generator model alone is capable of realistic multimodal synthesis, which does not require EBM revision (i.e., MCMC sampling) at test time. As shown in Sec. 5.2 and Sec. 5.5, the generator effectively “catches up’’ with the EBM, achieving strong multimodal generation quality in a **one-step, non-iterative manner**. This provides a test-time efficiency advantage similar to ECVAE.

---

### Official Review · Reviewer_zBpm · 2025-10-30

**Soundness:** 2
**Presentation:** 3
**Contribution:** 2
**Rating:** 4
**Confidence:** 4

**Summary:**

The paper presents a cooperative learning framework for multimodal generative modeling that unifies an energy-based model (EBM), a shared latent-variable generator, and a joint inference model within a single probabilistic system. The approach interleaves their maximum-likelihood updates with short-run MCMC refinements, allowing each component to benefit from the others. Specifically, the generator provides coherent multimodal initializations for EBM sampling, the inference model offers informative latent initializations for posterior sampling, and the EBM delivers corrective “revision signals” that refine both models. This self-correcting interaction mitigates poor MCMC mixing and ensures consistent cross-modal representations, enabling stable and efficient training. Experiments on PolyMNIST and CUB datasets demonstrate that the proposed multimodal EBM achieves superior synthesis quality and coherence, outperforming strong baselines such as diffusion-augmented CMVAE in FID and consistency metrics.

**Strengths:**

- This work proposes a cooperative framework that combines the complementary strengths of VAE and MCMC, effectively addressing both the limited expressivity of unimodal Gaussian-based VAEs and the slow mixing of noise-initialized MCMC.
- The method is evaluated across multiple benchmarks, which are familiar with the multimodal VAE community, demonstrating its applicability to various multimodal settings, including image–text and multimodal synthesis tasks.
- The ablation studies show the best recipit of hyperparameters. The results show that increasing the number of short-run MCMC steps from 10 to 30 yields clear improvements (with diminishing returns beyond 60).

**Weaknesses:**

- While the paper emphasizes the idea of combining VAE and MCMC in a cooperative manner, this concept itself has already been explored in prior studies [1, 2, 3]. The present work seems to be an incremental extension that adapts this idea to the multimodal setting.
- There is no comparison between your method and prior methods from the viewpoint of performance-time trade-off. MCMC converges slowly, so I doubt that Diff-CMVAE is superior from that viewpoint.
- It is understandable that text–image consistency is not typically evaluated for CUB in multimodal VAE research, but recent tools such as CLIP make it possible to measure text–image similarity. Could the authors also report a similar trade-off curve for CUB, as they did for PolyMNIST, to illustrate the balance between fidelity and consistency?
- The experiments mainly focus on PolyMNIST and CUB, which are relatively small datasets drawn primarily from the multimodal VAE literature. I wonder whether your method works well on datasets that contain larger images or modalities beyond text and image.
- Although the proposed method outperforms Diff-CMVAE on CUB in FID, there is no direct comparison with recent, more powerful diffusion-based text-to-image models. How does this method position itself when compared to modern diffusion models? The authors claim superiority in terms of FID and modality consistency, but additional analysis is needed to clarify its significance in the broader SOTA landscape.

[1] Hoffman, et al. Learning Deep Latent Gaussian Models with Markov Chain Monte Carlo. ICML2017.
[2] Taniguchi, et al. Langevin Autoencoders for Learning Deep Latent Variable Models. NeurIPS2022.
[3] Grathwohl, et al. No MCMC for me: Amortized sampling for fast and stable training of energy-based models. ICLR2021.

**Questions:**

Please see the weakness.

---

> ### Author Response · Authors · 2025-11-14
>
> We sincerely thank the reviewer for the valuable and detailed feedback. Please find our answers in **general responses**. Below, we address the remaining questions.
>
> ### Discussion with [1,2,3].
>
> We appreciate the reviewer’s helpful references. Prior studies [1, 2] explore MCMC posterior sampling for the generator model, while [3] introduces amortized EBM training through a complementary generator and inference model. However, all of these methods focus on single-modality learning. In contrast, our work develops a cooperative multimodal framework that jointly learns a multimodal EBM, shared generator, and joint inference model. This design specifically addresses challenges unique to multimodal settings, such as (i) modeling the shared inter-modal dependencies among heterogeneous modalities, and (ii) providing consistent initialization and MCMC revision across both data space and latent space. A more detailed theoretical and methodological comparison is provided in **general responses G.1 and G.2**.
>
> ### Challenging Benchmark and SOTA comparison.
>
> We thank the reviewer for recognizing the importance of scalability and strong baselines. As presented in Appendix A.1, we conduct experiments on high-resolution and large-scale benchmarks, demonstrating that our framework scales effectively to larger and more complex datasets. As noted by **Reviewer** **qamK**, these large-scale experiments confirm the robustness of our proposed learning scheme, while variational multimodal models often face difficulty scaling to such settings. We emphasize that this work is **not intended to pursue SOTA performance**, but rather to introduce a **novel, principled learning framework** that enables efficient and coherent multimodal framework. Our comprehensive experiments demonstrate clear advantages in coherence, stability, and efficiency, validating the strength of the proposed approach.

---

> ### Author Response · Authors · 2025-11-26
>
> We hope our responses above address your comments. We sincerely appreciate your thoughtful insights, and we are happy to elaborate further on any point that can be helpful.

---

> > ### Comment · Reviewer_zBpm · 2025-11-26
> >
> > While I appreciate the authors’ explanation, I would like to clarify my remaining concerns.
> >
> > Your method indeed attempts to (i) model the shared inter-modal dependencies among heterogeneous modalities, and (ii) provide consistent initialization and MCMC refinement in both data space and latent space. However, the technical foundations of these components have already been proposed in single-modality models. Extending these ideas to the multimodal setting, without introducing fundamentally new principles, feels incremental and limited in scope.
> >
> > Furthermore, if the proposed framework cannot scale to large settings such as text–image or text–video modeling, the case for building on variational multimodal backbones becomes less compelling.
> >
> > For these reasons, I remain unsatisfied with the level of novelty and impact, and therefore, I will keep my initial score.

---

> ### Author Response · Authors · 2025-11-26
>
> We thank the reviewer for the continued engagement with our work. We clarify our perspective on novelty and impact below.
>
> ### On the novelty of cooperative ideas to multimodal learning
>
> You are correct that the components of EBMs, generators, and inference models have been studied in single-modality settings. However, upon these components, our goal is to develop a unified cooperative framework that addresses challenges unique to *multimodal* learning.
>
> - **Inter-modal dependencies:** Unlike single-modality settings, multimodal data introduce a structured coupling (inter-modal relationships) that cannot simply be factorized or treated independently. Our framework incorporates this coupling directly into both the EBM and generator learning objectives.
> - **MCMC revision for capturing inter-modal dependencies**: In the single-modal case, the data-space EBM and generator-posterior MCMC are interdependent. While, in multimodal data, because each modality contributes partial information about the shared latent variable, ensuring coherent initialization and refinement across modalities is substantially more complex than in single-modality cooperative training.
> - **Unified probabilistic framework with multimodality:** The introduction of multimodal *shared generator model* and *joint multimodal inference model* within the cooperative loop fundamentally changes the learning dynamics and allows multimodal initialization for both data-space and latent-space MCMC, which does not exist in the single-modality literature.
>
> We hope to emphasize that the multimodal case fundamentally changes both the problem structure and the nature of cooperative learning. This required re-deriving the learning dynamics (**Sec. 3**) and constructing a multimodal generator and joint inference model that accounts for inter-modal dependencies. We provide empirical analysis in **Sec.5.3** and **Sec.5.4** that directly applying single-modality cooperative methods to the multimodal setting leads to ineffective learning dynamics and unstable energy estimation.
>
> In contrast, with our crucial adaptation, we are able to (i) stabilize the multimodal EBM learning, (ii) ensure coherent refinement across data and latent spaces, and (iii) achieve significantly improved multimodal consistency. This demonstrates that the multimodal extension is not merely an extension, but introduces essential modeling and algorithmic components required for successful multimodal generative learning.
>
>
>
> ### Large-scale dataset challenge
>
> In the Appendix. A.1 and A.2, we train our method on the high-resolution 256x256 CUB text-image dataset and the large-scale MSCOCO text-image dataset. In addition, we provide the comparison of generation quality in Tab.B.
>
> | FID           | Ours  | MMVAE+ |
> | ------------- | ----- | ------ |
> | CUB (256x256) | 56.32 | 213.74 |
> | MSCOCO        | 68.94 | 187.22 |
>
> Note that, for fair comparison, we use **exactly the same** shared generator and joint inference network and evaluate the synthesis drawn from our generator model (i.e., without MCMC EBM sampling for synthesis). The results demonstrate our cooperative framework can outperform our baseline by a wide margin, suggesting the effectiveness of our framework beyond small benchmarks and providing clear benefits over prior multimodal arts.
>
> We have updated our manuscript accordingly. We hope these answers help clarify the concerns about our method.

---

### Official Review · Reviewer_bVv4 · 2025-10-30

**Soundness:** 3
**Presentation:** 4
**Contribution:** 2
**Rating:** 6
**Confidence:** 4

**Summary:**

The paper presents a generative model for problems in which instances come in different modalities, e.g. text and images. The paper is rather dense and brings together a number of concepts. The authors did a very good work in presented their model in a comprehensive manner, given a bit of patience.

I will try to summarize as best as I can what the model is about. To do so I will use as a starting point a basic VAE model that operates on the multimodal setting, instances are structured as $\bf X = (\bf x_1, ..., \bf x_M )$. The latent variable $\bf z$ will be shared across all modalities. We then have:

* the encoder $q_\phi(\bf z | \bf X) = \sum_i q_{\phi_i} (z| \bf x_i)$  where the modality specific learnt posteriors are modelled as Gaussians $q_{\phi_i} ({\bf z}|  {\bf x_i} )=\mathcal N(\mu_{\phi_i}({\bf x_i}), \Sigma_{\phi_i}({\bf x_i}) )$ making  $q_\phi({\bf z} | {\bf X})$ a mixture of Gaussians.
* the joint latent model is: $p_\omega({\bf X}, {\bf z}) = p_\omega({\bf X} | {\bf z}) p_0({\bf z}) = \prod p_{\omega_i}( {\bf x_i} | {\bf z} ) p_0({\bf z})$, where again $p_{\omega_i}( {\bf x_i} | {\bf z}) = \mathcal N(\mu_{\omega_i}({\bf z}), \sigma )$

The authors remark that the particular parametrization of the inferred posterior lacks representation power for the multimodal setting, making it a not very appropriate modelling choise for the posterior. An alternative would have been to use Langevin Dynamics (LD) to sample from the posterior, $p_\omega({\bf z} | {\bf X }) \propto p_\omega( {\bf X | \bf z}) p_0( \bf z ) = \prod p_{\omega_i}( {\bf x_i | \bf z}) p_0( \bf z ) $, but here too there are issues such as poor mixing and small convergence when the prior $p_0( \bf z ) $, is not very informative, in addition the product decomposition requires that all modalities are present.

The authors propose:
1. to improve upon the representation power of the inferred posterior ${\bf z}' \sim q_\phi(\bf z | \bf X)$ by following it up with LD to sample from the posterior  $p_\omega({\bf X} | {\bf z})$ where the $q_\phi(\bf z | \bf X)$ will provide a more informed prior for the LD sampling. They denote the result of this completion/refinement by $M_{\omega}^{k_z}q_{\phi}({\bf z}|{\bf X})$.
2. They similarly complete the generation part ${\bf X}' \sim p_\omega( {\bf X | \bf z}) p_0( \bf z )$ by using the sampled ${\bf X}'$ to initialise one more LD sampling process this time over an energy-based model, $\pi_\alpha(.)$, on $\bf X$. They denote this completion by $M_{\alpha}^{k_x}q_{\omega}({\bf X})$; this produces a richer model than the one operating solely on the product of Gaussians ($p_\omega({\bf X} | {\bf z}) p_0({\bf z}) = \prod p_{\omega_i}( {\bf x_i} | {\bf z} ) p_0({\bf z})$).

Thus their model will learn:
* the latent inference model $q_\phi(\bf z | \bf X)$
* the joint generative model $p_\omega({\bf X} | {\bf z})$
* and the energy model $\pi_\alpha( {\bf X} )$

each of these components produce KL terms towards the construction of the final loss function. The authors nicely explain the effect of each KL term and the interplay of the different components of the model.

The evaluation takes place in two datasets, a multimodal version of MNIST, and a bimodal dataset with a text and an image modality. The authors compare against a large number of baselines, most of them being variants/extensions of the simplified VAE structure given above. They evaluate the coherence of the model over the different modalities, namely whether the different modalities agree semantically, e.g. same digit in PolyMNIST over the modalities, and generation quality as quantified by FID.

**Strengths:**

* Well written paper with a careful analysis and explanations of the contribution of the different components.
* Results seem to indicate meaningful improvements over the baselines considered.

**Weaknesses:**

* Computational complexity, since within training we add two MCMC procedures.
* Somehow limited novelty with respect to the previous variants, namely use the learnt components as more informative priors in the two MCMC procedures, nevertheless they do the work.
* Baselines mostly revolve around variational models which perform rather badly compared to the proposed method and a Diffusion enhanced VAE.

**Questions:**

* How long does it take to train, in table 1 you give a 1 to 3 seconds / iteration; what should I understand by iteration here? one batch update? what is the computational overhead compared to switching off the MCMC procedures?
* For the conditional experiments I am not sure I saw much information. What is the conditioning factor? digit id in the case of MNIST or one of the modalities? and what in the case of CUB? on which parts of the model the condition is added? Or is it that one of the modalities is used to produce the latent variable $\bf z$ by sampling from the respective inferred posterior $q_{\phi}({\bf z} | {\bf x_i})$
* On the conditional generation results visualisation (figure 8) were the samples produced by the authors code or copied from Palumbo 2023? In the latter case where all the architectural details equivalent? would have been also interesting to have the visualisations of (Diff)-CMVAE.
* Numerical values FID on PolyMNIST? is this what is shown in table 6 of the appendix? Also in figure 1 each model appears three times, are these different trained models and their respective performances. And why this difference in the presentation of results (FID) table for CUB, scatter plot for PolyMNIST?
* How is coherence evaluated in the case of CUB? or isn't evaluated? I am not sure I saw something like that whether in the main text or in the appendix
* Ι would be curious to know how the proposed method operates in standard single modality settings compared to a baseline in which the MCMC is switched off; I guess this is probably explored in past work.

---

> ### Author Response · Authors · 2025-11-14
>
> We sincerely thank the reviewer for the valuable and supportive feedback. Most of the main concerns are addressed in the **general responses**, and we provide additional clarifications below.
>
> ### Visualization in Figure 8
>
> We employ **exactly the same network architectures** as our baselines and take the result from the original paper. We have also **updated our manuscript** to include the visualization of Diff-CMVAE in Figure. 8. Both methods produce realistic samples; however, our joint learning framework offers notable advantages as discussed in **general response G.3**.
>
> ### Numerical values on PolyMNIST
>
> In Fig. 1, we intend to provide a broad comparison over the landscape and thus plot their variants results of using different training configurations (e.g., varying the KL-divergence weight $\beta$ in our variational baselines). In Tab. 6, we report the numerical results corresponding to the **best-performing variant** of each baseline.

---

> ### Author Response · Authors · 2025-11-26
>
> We hope that our clarifications and updated results address your concerns. Thank you again for your constructive feedback, and please feel free to let us know if there is anything else we can clarify. We would be happy to provide further clarification.

---

### Official Review · Reviewer_fqno · 2025-11-02

**Soundness:** 2
**Presentation:** 3
**Contribution:** 2
**Rating:** 4
**Confidence:** 3

**Summary:**

This paper learns multimodal EBMs by interweaving MLE with short-run MCMC revisions across three components: an EBM, a shared generator, and a joint inference model, so the generator and inference networks act as complementary initializers for effective EBM sampling. Experiments on CUB and PolyMNIST show that the proposed method achieves better multimodal coherence/quality, and ablations highlight the contributions of the shared generator and joint inference.

**Strengths:**

- The paper is well organized and easy to follow: the problem setup, method, and objectives are laid out coherently, and the sectioning and notation make the technical ideas accessible.
- While prior multimodal VAEs with iterative methods have used diffusion-based methods to improve either generation or inference, this work is the first to make both inference and generation iterative and to feed both revisions back cooperatively into training. This dual, mutually reinforcing loop is a genuine novelty.

**Weaknesses:**

- Unclear global objective across the three losses. Section 3.1.1 specifies the objectives and §3.2.2 discusses their meanings, but it remains unclear what single global quantity the three optimizations jointly minimize/maximize. Consequently, it is not obvious why these particular objectives are necessary, and it seems plausible that other cooperative training objectives for heterogeneous models could be devised to similar effect.
- Limited and potentially unfair comparison to iterative baselines (e.g., Diff-CMVAE). Because the proposed method, like Diff-CMVAE, requires iterative procedures, comparisons should clearly control for this. However, the paper compares against Diff-CMVAE only on CUB conditional FID; for CUB unconditional metrics, for coherence, and for PolyMNIST, the comparisons are primarily to non-iterative baselines. The authors should include Diff-CMVAE (or comparable iterative multimodal diffusion methods) in these settings as well. In particular, claiming superior FID on PolyMNIST without reporting Diff-CMVAE’s FID is not fair.
- Lack of ablation-style cost–performance scaling vs. Diff-CMVAE. The method incurs iteration in both inference and generation; Diff-CMVAE iterates only for generation. It is required to quantify how runtime and accuracy scale (e.g., with latent size, image resolution, number of MCMC steps) relative to Diff-CMVAE, to clearly show the additional computation required and the resulting performance trade-offs.

**Questions:**

Please address the above concerns with concrete clarifications and, where appropriate, additional experiments.

---

> ### Author Response · Authors · 2025-11-14
>
> We sincerely thank the reviewer for the constructive and thoughtful comments. Please find our response in **general response**. We provide further clarification below.
>
> ### Global objective and joint learning.
>
> We appreciate this insightful observation. Indeed, our method is not derived from a single global objective, as the inclusion of finite-step MCMC makes it inherently *non-amortized*. Such global objectives can be derived by "switching off" MCMC, which resembles amortized methods.  However, compared to amortized methods, our MCMC-based cooperative framework demonstrates its unique advantages for both the multimodal modelling and learning dynamics. Please see more analysis in **generator response G.1, G.2**.
>
>
>
> ### Comparison and ablation with Diff-CMVAE and non-iterative methods.
>
> We agree that comparing iterative methods fairly is important. Diff-CMVAE introduces a second-stage diffusion model trained only in the image space for CUB, making it ad-hoc for multimodal scenarios: an $M$-modality dataset would require training $M$ separate diffusion models. Because each diffusion model operates independently, Diff-CMVAE **cannot** perform unconditional multimodal synthesis within a unified latent space, nor can it scale feasibly to datasets such as PolyMNIST (which would require five separate diffusion networks).
>
> For reported results, we use the same network structures (generator and inference model), latent size, image resolution as CMVAE. We also compare model complexity (EBM v.s. diffusion) and training cost. As summarized in **general response G.3** (Tab. A), our method achieves superior training and sampling efficiency.

---

> ### Author Response · Authors · 2025-11-26
>
> We hope that the additional analysis, comparisons, and clarifications above help address your concerns. We truly appreciate your careful evaluation of our work, and we would be glad to provide any further details or discussion that might be helpful.

---

### Author Response · Authors · 2025-11-14

We thank all reviewers for their valuable feedback. We are encouraged to hear that many of you found (1) our paper to be well-organized and clearly presented (**Reviewers** **fqno**,**bVv4**); (2) our idea to be novel (**Reviewer fqno**), meaningful (**Reviewer bVv4**), and effective in addressing existing limitations in the literature (**Reviewers zBpm, qamK**); and (3) our experiments and ablation studies to be a strong aspect of the work (**Reviewers fqno, bVv4, zBpm, qamK**). We shall carefully update our manuscript based on the discussions here.

Below, we provide **general responses** for common concerns and respond to each reviewer in their individual comments.

---

> ### Author Response · Authors · 2025-11-14
>
> ### G.1. Why we use MCMC-revision for multimodality?
>
> The core motivation of our work is to develop a **principled cooperative learning** framework that jointly trains a multimodal energy-based model (EBM), a shared latent generator, and a joint inference model. Learning multimodal EBMs through maximum likelihood estimation (MLE) is challenging because noise-initialized MCMC sampling often mixes poorly, which becomes even more severe in multimodal settings due to strong inter-modal dependencies among heterogeneous modalities. Conversely, learning the shared generator via variational inference, as in existing multimodal VAEs, is limited by the mismatch between the true generator posterior and the variational approximation (see analysis in Sec. 3.1), resulting in degraded multimodal coherence. While MCMC posterior sampling could in principle address this mismatch, its effectiveness is largely hindered when initialized from uninformed noise, leading again to poor mixing and unstable learning.
>
> Our framework addresses these **coupled challenges** by interleaving the MLE updates of all three components — EBM, generator, and inference model — through MCMC revisions. Specifically, (i) the generator provides coherent multimodal initializations for EBM MCMC sampling, and (ii) the inference model offers informative latent initializations for generator posterior MCMC sampling. The MCMC-revised samples, in turn, act as mutual revision signals that refine both the generator and inference models. This cooperative loop forms a unified probabilistic mechanism in which all three models reinforce one another, enabling efficient sampling, stable training, and coherent multimodal synthesis (see also **G.2** for learning objective analysis).

---

> ### Author Response · Authors · 2025-11-14
>
> ### G.2. Can we turn off MCMC sampling in the framework?
>
> Several recent advances have investigated EBM learning without explicit MCMC sampling [1, 2] (we thank **Reviewer zBpm** for highlighting [1]). These works study **single-modal** EBMs that employ amortized samplers to *replace* MCMC, thereby avoiding iterative sampling. In contrast, our focus is on the **multimodal setting**, which introduces two additional challenges: (i) effectively capturing the shared inter-modal relationships across heterogeneous modalities, and (ii) mitigating the mismatch induced by multimodal joint inference models. To address these challenges, we incorporate MCMC revision as a **key component** of our cooperative framework, which allows both the EBM and the generator posterior to be iteratively refined by each other, ensuring coherent multimodal alignment that **cannot be achieved by amortized single-pass updates**.
>
> Moreover, the inclusion of MCMC revision makes our **learning objectives fundamentally different** from previous “no-MCMC’’ formulations. For clarity, and to directly illustrate the difference in learning dynamics independent of modality notation, we denote $\Omega,\Phi$ as shorthand for MCMC-revised densities $\Omega_{\omega,\alpha}(\mathbf{X}, \mathbf{z}), \Phi_{\omega,\phi}(\mathbf{X},\mathbf{z})$  and denote $Q=q_\phi(\mathbf{z}|\mathbf{X})p_{data}(\mathbf{X})$, $\Pi=\pi_\alpha(\mathbf{X})q_\phi(\mathbf{z}|\mathbf{X})$,$P=p_\omega(\mathbf{X},\mathbf{z})$ for joint densities of amortized models. We denote "AM" for methods using amortized models without MCMC.
>
> - **Learning the EBM ($\alpha$):** The corresponding KL terms in our method (Eqn. 10) and AM are: $\min_\alpha KL(\Phi\| \Pi) - KL(\Omega\|\Pi)$ v.s. $\min_\alpha KL(Q\| \Pi) - KL(P\|\Pi)$. Our formulation leverages MCMC-revised samples (i.e., joint densities of $\Omega$ and $\Phi$), whereas AM relies solely on ancestral samples ($Q$ and $P$). Because our samples are refined by the EBM itself, they provide a more accurate approximation of the target energy landscape $KL(M_{\alpha_t}p_{\omega_t}(\mathbf{X})\|\pi_{\alpha_t}(\mathbf{X}))\le KL(p_{\omega_t}(\mathbf{X})\|\pi_{\alpha_t}(\mathbf{X}))$. This results in more effective and stable EBM learning and leverages the contextual modelling capability of EBM to effectively guide the multimodal generator model.
>
> - **Learning the (shared) generator model ($\omega$):** For learning the generator model (Eqn. 11), KL terms for ours and AM are $\min_\omega KL(\Phi\|P)+KL(\Omega\| P)$ v.s. $\min_\omega KL(Q\|P) + KL(P\| \Pi)$. The learning dynamics differ substantially. In our case, the generator is trained with MCMC-revised latent samples, yielding a closer match to the true generator posterior: $KL(M_{\omega_t}q_{\phi_t}(\mathbf{z}|\mathbf{X})\|p_{\omega_t}(\mathbf{z}|\mathbf{X}))\le KL(q_{\phi_t}(\mathbf{z}|\mathbf{X})\|p_{\omega_t}(\mathbf{z}|\mathbf{X}))$, which aims to address the mismatch between the generaetor posterior and joint inference model (analysis in Sec. 3.1). In addition, $KL(\Omega\|P)$ learns to match the revised MCMC samples from EBM-refined samples, while the Amortizer method intends to chase the major modes of $\pi_\alpha(\mathbf{X})$ through variational approximation (i.e., $KL(P\| \Pi)$). Hence, our generator directly learns from revised multimodal samples that can better capture inter-modal consistency.
>
> - **Learning the (joint) inference model ($\phi$).** For the inference model (Eqn. 12), learning objectives for ours and AM are: $\min_\phi KL(\Phi\| Q) + KL(\Omega\|\Pi)$ v.s. $\min_\phi KL(Q\|P) + KL(P\|\Pi)$. The two approaches differ in both learning source and optimization target. Our inference network amortizes latent MCMC refinement on observed data (i.e., $KL(\Phi\| Q)$), while AM performs pure variational inference (i.e., $KL(Q|P)$). On generated samples, our model matches EBM-revised generator samples (i.e., $KL(\Omega|\Pi)$), whereas AM directly uses ancestral generator outputs ($KL(P|\Pi)$), which can lead to sub-optimal inference quality.

---

> ### Author Response · Authors · 2025-11-14
>
> ### G.3. What MCMC-revision incurs and benefits compared to Diff-CMVAE?
>
> Our proposed learning algorithm belongs to the family of MCMC-based cooperative methods, and we acknowledge that it introduces computational overhead due to its iterative nature compared to purely variational frameworks. We report in Tab. 1 where training time per iteration refers to one minibatch update.
>
> To further provide a comprehensive comparison, we additionally report the total training time, parameter complexity, and sampling time in Tab. A (to be included in the revision). In addition, we also compute **CLIP score** for CUB coherence evaluation (thanks to the suggestion of **Reviewer zBpm**). For fairness, we reproduce the Diff-CMVAE results using the official implementation from [3, 4] and follow their original experimental setup. All results are reported using a single Nvidia A100 40GB GPU with the same experiment settings.
>
> **Tab.A** Comparison for training cost, sampling time, FID ($\downarrow$), and CLIP scores ($\uparrow$).
>
> |                                  |           Ours           |  CMVAE  |               Diff-CMVAE                |  MMVAE  |
> | -------------------------------- | :----------------------: | :-----: | :-------------------------------------: | :-----: |
> | Total Training Time (300 epochs) |         2.2 day          | 1.2 day | 1.2 (CMVAE) + 3.1 (Diffusion) = 4.3 day | 1.0 day |
> | Sampling Time (second / batch)   |  0.001(Gen)/0.08 (EBM)   |  0.001  |                  38.12                  |  0.001  |
> | FID (T2I)                        | 26.15 (Gen)/25.98 (EBM)  | 155.11  |                  28.00                  | 232.20  |
> | FID (uncond)                     | 21.45 (Gen)/20.72 (EBM)  | 141.00  |                   N/A                   | 213.89  |
> | CLIP (uncond)                    | 0.280 (Gen)/0.284 (EBM)  |  0.263  |                   N/A                   |  0.231  |
> | CLIP(T2I)                        | 0.278 (Gen)/0.282 (EBM)  |  0.260  |                  0.272                  |  0.242  |
> | CLIP(I2T)                        | 0.286 (Gen) /0.290 (EBM) |  0.272  |                   N/A                   |  0.235  |
> | Parameter Overhead               |     28,782,464 (EBM)     |    /    |        141,470,979 (Diff U-Net)         |    /    |
>
> It is important to note that Diff-CMVAE follows a two-stage training pipeline, where the diffusion model is separately trained and applied only on the image space of CUB. Consequently, for an $M$-modality dataset, $M$ independent diffusion models must be trained—making it an ad-hoc and modality-specific design rather than a unified multimodal framework. In contrast, our method jointly learns the multimodal EBM, shared generator, and joint inference model. To sum up, our MCMC-revision provides substantial benefits:
>
> - **Joint optimization:** The generator and inference model are learned cooperatively and jointly under EBM guidance, without requiring per-modality diffusion models.
> - **Faster and more scalable training:** Despite iterative updates, our total training time is shorter than Diff-CMVAE with a smaller model size, and it does not require modality-specific second-stage adaptation.
> - **Efficient sampling:** Our generator **alone** (see **Tab.A**) can synthesize realistic multimodal samples (FID = 26.15 on CUB) in *one step,* i.e., non-iterative manner (**Reviewer fqno**), whereas Diff-CMVAE generator cannot be reused independently for high-quality synthesis, since its diffusion model is trained separately, which requires 250 diffusion sampling steps.

---

> ### Author Response · Authors · 2025-11-14
>
> ### G.4. Inference under Missing Modalities (Reviewer bVv4, qamK)
>
> Our framework follows the standard inference mechanism established in multimodal VAEs for handling missing modalities. Given any available modality $\mathbf{x}_i$, we first infer its shared latent variable, and then use this latent variable to generate the missing modalities through the shared generator for $\mathbf{x}_j$ where $j \ne i$. This mechanism is identical to prior multimodal VAE baselines, ensuring fair comparison and consistent inference behavior.
>
> More importantly, our method extends this process by introducing MCMC refinement on the generator posterior, which is infeasible for purely variational models. Specifically, after obtaining the initial inferred latent variable, we apply a few steps of posterior Langevin dynamics for refinement using all (if available) additional modalities (as reported in Sec. A.3). This refinement enhances cross-modal consistency and allows the model to recover more coherent missing modalities.
>
> In our experiments, we evaluate using the same inference mechanism as in baseline models to ensure fairness, and the results consistently show superior reconstruction and coherence. The optional MCMC refinement further improves consistency across modalities without requiring any change in the baseline inference pipeline.
>
>
>
> [1] Grathwohl, et al. No MCMC for me: Amortized sampling for fast and stable training of energy-based models. ICLR 2021.
>
> [2] Han, et al. Divergence triangle for joint training of generator model, energy-based model, and inferential model. CVPR 2019.
>
> [3] Palumbo, et al. Deep Generative Clustering with Multimodal Diffusion Variational Autoencoders. ICLR 2024
>
> [4] Pandey, et al. Diffusevae: Efficient, controllable and high-fidelity generation from low-dimensional latents. TMLR 2022

---

### Meta-Review · Area_Chair_Qbux · 2026-01-06

**Summary:**

This paper proposes a learning algorithm for multimodal EBM that uses maximum likelihood updates and a short run of a Markov chain.
Reviewer raised several concerns about the novelty of the proposed algorithm and missing baselines/ablation studies. The authors partially addressed concerns by providing additional experimental results in their response. However, I think the most critical concern about the novelty still remains since the main idea of cooperatively using VAE and MCMC has been used in several prior works. While the authors argue the challenges in modelling their algorithm for multimodal learning, it seems that the proposed method is a minor extension of existing ideas. Hence, I recommend the rejection of this paper.

**Reviewer Concerns:**

I think the following concerns are not fully addressed:
- On the novelty of the method (Reviewer fqno). While the authors argue the challenges in designing their method, the main idea of cooperatively using VAE and MCMC has already been used in prior works.
- The function of objective functions (Reviewer fqno). The authors did not describe their global objective involving MCMC and why particular objectives are necessary in their response.

**Reviewer Scores:**

I do not think the reviewers to change their scores during the discussion period.

---

### Decision · Program_Chairs · 2026-01-26

Reject